# Faster decline and higher variability in the sea ice thickness of the marginal Arctic seas when accounting for dynamic snow cover

Robbie D.C. Mallett [1], Julienne C. Stroeve [1,2,3], Michel Tsamados [1], Jack C. Landy [4,5], Rosemary Willatt [1], Vishnu Nandan [3], and Glen E. Liston [6]

[1]Centre for Polar Observation and Modelling, Earth Sciences, University College London, London, UK
[2]National Snow and Ice Data Center, University of Colorado, Boulder, CO, USA
[3]Centre for Earth Observation Science, University of Manitoba, Winnipeg, Canada
[4]School of Geographical Sciences, University of Bristol, Bristol, UK
[5]Department of Physics and Technology, UiT The Arctic University of Norway, Tromsø, Norway
[6]Cooperative Institute for Research in the Atmosphere, Colorado State University, Fort Collins, CO, USA

**Correspondence:** Robbie Mallett (robbie.mallett.17@ucl.ac.uk)

**Abstract.** Mean sea ice thickness is a sensitive indicator of Arctic climate change and in long-term decline despite significant interannual variability. Current thickness estimations from satellite radar altimeters employ a snow climatology for converting range measurements to sea ice thickness, but this introduces unrealistically low interannual variability and trends. When the sea ice thickness in the period 2002-2018 is calculated using new snow data with more realistic variability and trends, we find mean sea ice thickness in four of the seven marginal seas to be declining between 60-100% faster than when calculated with the conventional climatology. When analysed as an aggregate area, the mean sea ice thickness in the marginal seas is in statistically significant decline for six of seven winter months. This is observed despite a 76% increase in interannual variability between the methods in the same time period. On a seasonal timescale we find that snow data exerts an increasingly strong control on thickness variability over the growth season, contributing 46% in October but 70% by April. Higher variability and faster decline in the sea ice thickness of the marginal seas has wide implications for our understanding of the polar climate system and our predictions for its change.

## 1 Introduction

Sea ice cover moderates the exchange of moisture, heat and momentum between the atmosphere and the polar oceans, influencing regional ecosystems, hemispheric weather patterns and global climate. Sea ice thickness (SIT) is a key characteristic of the sea ice cover, as thicker sea ice weakens the coupling between the ocean and atmosphere systems.

Thicker sea ice is more thermally insulating and limits heat transfer from the ocean to the atmosphere in winter and consequent thermodynamic growth (Petty et al., 2018a). SIT also exerts control on sea ice dynamics and rheology (Tsamados et al., 2013; Vella and Wettlaufer, 2008). The thickness of sea ice during snow accumulation also dictates whether the sea ice surface drops below the waterline, potentially increasing thermodynamic sea ice growth through the formation of snow-ice (Rösel et al., 2018). The impact of the end-of-winter SIT distribution persists into the melt season with thick sea ice decreasing the transmission of solar radiation to the surface ocean and reducing the potential for in- and under-ice primary productivity

(Mundy et al., 2005; Katlein et al., 2015). Finally, thick sea ice is far more likely to survive the melt season, increasing the average age of Arctic sea ice. Correct assimilation of ice thickness into models therefore offers opportunities for prediction of the sea ice state on seasonal timescales (Chevallier and Salas-Mélia, 2012; Blockley and Peterson, 2018; Schröder et al., 2019).

The annual sea ice thickness distribution is highly spatially variable, with a cover of thick multi-year ice in the Central Arctic and a thinner, more seasonally variable cover of first year ice in the marginal seas. Regional sea ice thickness distributions are often characterised by the mean thickness, $\overline{SIT}$. As well as being a key parameter for the processes described above, the value can be multiplied by the sea ice area to produce the sea ice volume, one of the most sensitive indicators of Arctic climate change (Schweiger et al., 2019).

While continuous and consistent monitoring of Pan-Arctic SIT has not been achieved on a multi-decadal timescale, a combination of different techniques has suggested a significant decline in thickness since 1950 (Kwok, 2018; Stroeve and Notz, 2018). Satellite altimeters using both radar and lidar have provided a valuable record of changing sea ice thickness, but have often been limited for various reasons. Some have been limited spatially by their orbital inclination (e.g. the European Remote Sensing (ERS) satellites, Envisat, AltiKa and Sentinel radar altimeters have operated up to only 81.5 degrees north), and

others in temporal coverage (e.g. ICESat was operated in 'campaign mode' rather than providing continuous coverage). Two satellite altimeters currently offer continuous and meaningfully Pan-Arctic monitoring of the Arctic sea ice: the ICESat-2 and CryoSat-2 altimeters. ICESat-2 has been in operation since September 2018 and so far has documented only two winters of sea ice thickness (Kwok et al., 2020).

Although the launch of the CryoSat-2 radar altimeter (henceforth CS2) in 2010 allowed significant advances in understanding

the spatial distribution and interannual variability of Pan-Arctic SIT (Laxon et al., 2013), a statistically significant decreasing trend within the CS2 observational period has not been detected for the Arctic as a whole. The lack of certainty regarding any trend in SIT is in part due to the various uncertainties associated with SIT retrieval from radar altimetry (Ricker et al., 2014; Zygmuntowska et al., 2014). Major contributors to these uncertainties are the relatively large footprint of a radar pulse when compared to laser altimetry, the variable density of sea ice, retracking of radar returns from rough sea ice, and the need for an

*a priori* snow depth and density distribution (Kern et al., 2015; Landy et al., 2020).

The impact of snow-depth uncertainty on SIT retrievals was recently included by the IPCC in a list of 'Key Knowledge Gaps and Uncertainties' (Meredith et al., 2019). More specifically, Bunzel et al. (2018) found snow to have a strong influence on the interannual variability of SIT and consequent detection of thickness trends. Here we investigate the impact of a new, Pan-Arctic snow depth and density data set (SnowModel-LG; Liston et al., 2020; Stroeve et al., 2020) on trends and variability

in regional $\overline{SIT}$ when used in place of the traditional, climatological data set (Warren et al., 1999). We show that traditional calculations of $\overline{SIT}$ omit significant interannual variability due to their reliance on a snow climatology, and we quantify this omission. We also show that sea ice is likely thinning at a faster rate in some marginal seas than previously thought, because the snow water equivalent on the sea ice is declining too.

## 1.1 The Role of Snow in Radar-Altimetry Derived Sea Ice Thickness Retrievals

Satellite radar altimetry involves the emission of radar pulses from a satellite and the subsequent detection of their backscatter. The time difference between emission and detection ('time of flight') corresponds to the distance traveled and thus the height of the transmitter above the scattering surface. Radar altimeters of different frequencies have been carried on board several earth observation satellites such as ERS-1/2, Envisat, AltiKa, CryoSat-2 and Sentinel-3A/B (Quartly et al., 2019). We now quantify the role of snow cover in conventional sea ice thickness estimation, before revealing and explaining the effects of previously

unincorporated trends and variability.

The Ku-band radar waves emitted from CryoSat-2 are generally assumed in mainstream SIT products to scatter from the snow/sea-ice interface (Kurtz et al., 2014; Tilling et al., 2018; Hendricks and Ricker, 2019; Landy et al., 2020). The difference in radar ranging (derived from time-of-flight) between areas of open water and areas of sea ice is known as the 'radar freeboard', $f_r$. The height of the sea ice surface above the waterline is referred to as the sea ice freeboard, $f_i$. This is extracted from the

radar freeboard through (a) assuming that the primary scattering horizon corresponds to the sea ice surface, and (b) accounting for the slower radar wave propagation through the snow cover above the sea ice surface (Armitage and Ridout, 2015; Mallett et al., 2020). The sea ice freeboard can then be converted to sea ice thickness by considering the floe's hydrostatic equilibrium given the sea ice density and weight of overlying snow. In the simplified case of bare sea ice, we would calculate:

$$SIT_{bare} = f_r \frac{\rho_w}{\rho_w - \rho_i} \tag{1}$$

Where $\rho_w$ is the density of seawater and $\rho_i$ the density of sea ice. In order to adjust the above equation for the presence of overlying snow, the twin effects of the snow's weight and the snow's delaying influence on radar pulse propagation must be taken into account. These three influences on SIT (the radar freeboard measurement, the pulse propagation delay and the freeboard depression from snow weight) can therefore be expressed as three terms (see supplementary information) in the following way:

$$SIT = h_r \frac{\rho_w}{\rho_w - \rho_i} + h_s \frac{\rho_w}{\rho_w - \rho_i} \left[ \frac{c}{c_s} - 1 \right] + h_s \frac{\rho_s}{\rho_w - \rho_i} \tag{2}$$

In this manuscript we introduce a simple method for combining the second and third terms of the above equation into a single term that is proportional to the snow water equivalent (Sect. 3.1). This helps to easily separate the influences of snow data and radar freeboard measurements on the determination of sea ice thickness. Specifically, we compare the impact of two snow products on regional trends and variability in sea ice thickness. These products are the snow climatology given by Warren

et al. (1999) and the output of SnowModel-LG (Liston et al., 2020; Stroeve et al., 2020).

## 2 Data Description

### 2.1 Regional Mask

We define seven regions of the Arctic Basin using the mask from Stroeve et al. (2014) which is gridded onto a 25 km resolution EASE grid (Brodzik et al., 2012, Fig. 1). We define the 'marginal seas' of the Arctic Basin as the color coded areas of Fig. (1) excluding the Central Arctic. All constituent regions of the 'marginal seas' grouping lie within the coverage of Envisat barring a negligible portion of the Laptev Sea.

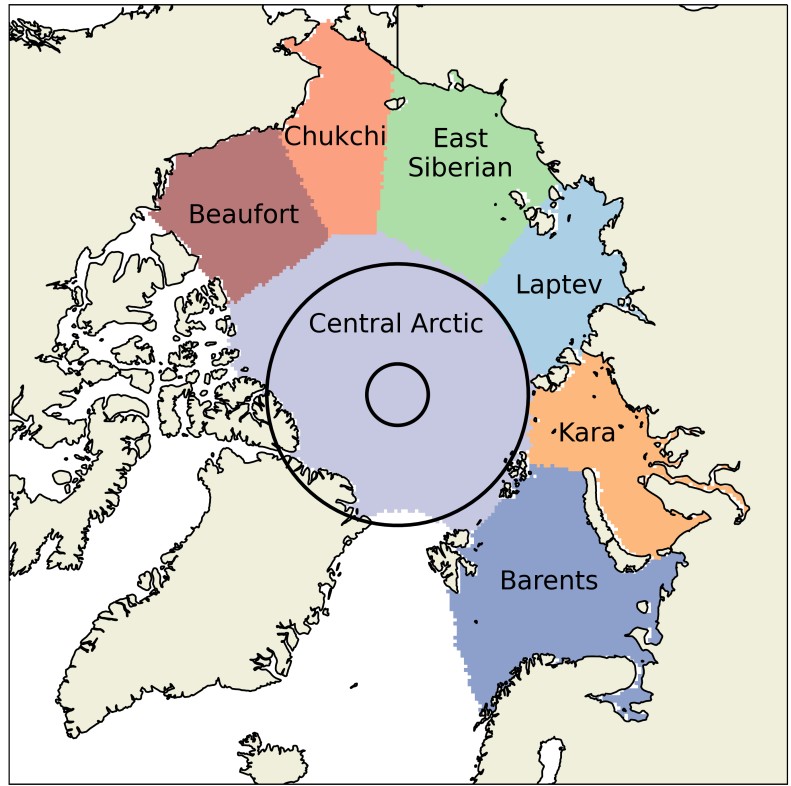

**Figure 1.** The definitions of the marginal Arctic seas used in this paper, from Stroeve et al. (2014). Two black, concentric circles indicate the latitudinal limits of the CryoSat-2 (inner circle; 88°N) and Envisat (outer circle; 82.5°N) missions.

### 2.2 Radar Freeboard Data

To examine the impact of snow products on Enivsat/CryoSat-2 thickness retrievals, we used radar freeboard data from the ESA Sea Ice Climate Change Initiative (Hendricks et al., 2018). This data is available from October in the winter of 2002/03 until April in the winter of 2016/17. This product was chosen for two main reasons: (a) it provides a consistent record for both the Envisat and CS2 missions (Paul et al., 2018), and (b) it is publicly available for download. CS2 carries a delay-Doppler

altimeter that significantly enhances along-track resolution by creating a synthetic aperture. For this reason as well as its higher latitudinal limit, we used CS2 radar freeboard measurements over Envisat during the period when the missions overlapped (November 2010 - March 2012). To create a radar freeboard product that is consistent between the Envisat and CS2 missions,

Envisat returns are retracked using a variable threshold retracking algorithm. This variable threshold is calculated from the strength of the surface backscatter and the width of the leading edge of the return waveform such that the inter-mission bias is minimised (Paul et al., 2018). The results are comprehensively analysed in the Product Validation & Intercomparison Report (ESA, 2018). One key finding of this report is that while Envisat radar freeboards are calculated so as to match CS2 freeboards during the period of overlap over the whole Arctic basin, there are biases over ice types. In particular, Envisat ice freeboards

(not radar freeboards) are biased 2-3 cm low (relative to CS2) in areas dominated by multi-year ice (MYI), and 2-3 cm high in areas dominated by first-year ice (FYI). We discuss the implications of these biases in Sect. 5.3.

While the ESA CCI data are only available from the CCI website until the winter of 2016/17, the CryoSat-2 radar freeboards in this data are identical to the CS2 radar freeboard product of the Alfred Wegener Institute (Hendricks and Ricker, 2019, this was manually confirmed). We were therefore able to extend our radar freeboard timeseries through the winter of 2017/18,

which is when our snow data from SnowModel-LG (see below) ends.

All radar freeboard data used in this study are supplied on a 25 km EASE grid (Brodzik et al., 2012), the same as that of SnowModel-LG.

## 2.3 The Warren Climatology (W99)

The most commonly used radar-altimetry SIT products use algorithms developed by the Centre for Polar Observation and

Modelling, the Alfred Wegener Institute and the NASA Goddard Space Flight Centre (Tilling et al., 2018; Hendricks and Ricker, 2019; Kurtz et al., 2014). Another commonly used but not publicly available product is from the NASA Jet Propulsion Laboratory (Kwok and Cunningham, 2015). All four groups utilize modified forms of the snow climatology assembled by Warren et al. (1999) from the observations of Soviet drifting stations between 1954 and 1991 (henceforth referred to as W99).

While the consistent use of W99 for sea ice thickness calculation is convenient for intercomparison of products (e.g. Sallila

et al., 2019; Landy et al., 2020), the data have a number of drawbacks. This work is centered around two key issues with the use of W99 for SIT retrieval: inadequate representation of interannual variability and trends.

The Warren Climatology includes quadratic fits for every month of snow water equivalent and snow depth. We projected these fits over the 361×361 EASE grid (for combination with our radar freeboard data and comparison with SnowModel-LG) to create SWE and depth distributions across the Arctic basin as defined in Sect. (2.1).

### 2.3.1 Drifting Station Coverage Illustration

At this point it is instructive to briefly illustrate the coverage of the drifting stations from which W99 was compiled. We analysed position and snow depth data from the twenty-eight drifting stations that contributed to W99 (Fig. 2a). It is clear that the vast majority of these operated in the Central Arctic or in the East Siberian Sea, with very little sampling done in most other marginal seas. But while these tracks illustrate the movements of the drifting stations, it is important to note that the stations

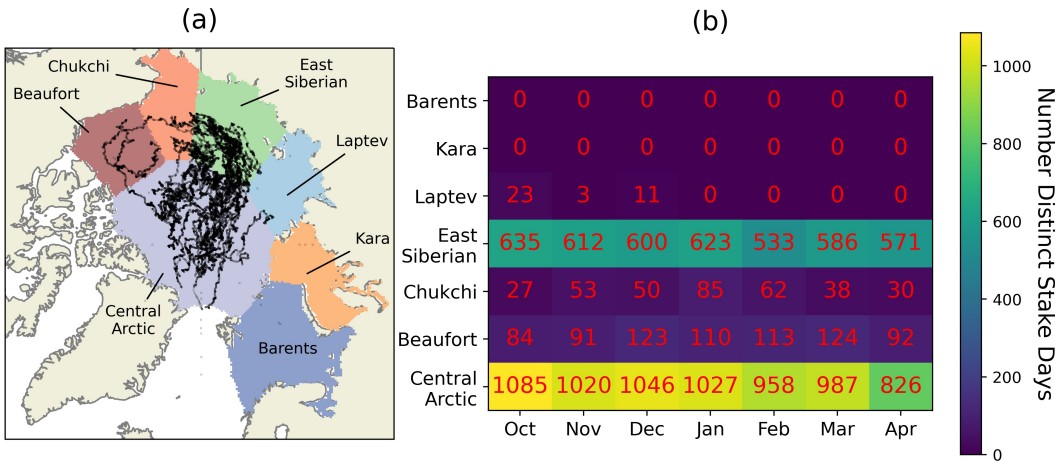

**Figure 2.** (a) tracks of Soviet drifting stations 3 - 31. (b) Number of days in each region in each month that snow stake measurements were taken.

were not always collecting snow data which would contribute to the W99 climatology. To assess the spatial distribution of snow sampling, we cross-referenced the position data with days on which the drifting stations recorded the snow depth at their measuring stakes. We then calculated the number of 'measurement-days' in each region-month combination (Fig. 2b). We note that when two drifting stations were operating at the same day, we count this as two distinct days (as they were rarely so close together so as to collect redundant data).

This reveals that no snow measurements were taken in the Barents and Kara Seas, and none in the Laptev Sea for four of the seven winter months. While 'snow-line' transect data also contributed to W99 (and indeed was used in preference to stake data where possible), we find that snow-line data was overwhelmingly collected on days where stake-data was also collected.

Figure 2 illustrates that the quadratic fits of W99 are not obviously appropriate for use in several of the marginals seas. However we note that a number of authors have still used the climatology for sea ice thickness retrievals in these regions, often in the course of sea ice volume calculations (e.g. Laxon et al., 2003, 2013; Tilling et al., 2015, 2018; Kwok, 2018; Li et al., 2020a; Belter et al., 2020; Li et al., 2020b). We therefore consider these regions in this manuscript, but with the understanding that mW99 is potentially not representative of the snow conditions.

### 2.4 The modified Warren Climatology (mW99)

The W99 climatology is by definition invariant from year to year, and was implemented in this way by Laxon et al. (2003) and Giles et al. (2008a) to estimate sea ice thickness using ERS 1 & 2. When implemented like this, the amount of snow on sea ice exhibits no interannual variability.

The implementation of W99 was then modified by Laxon et al. (2013) based on the results of Operation IceBridge flights which showed reduced snow depth over first year ice (Kurtz and Farrell, 2011). This implementation, known as 'mW99',

consists of halving snow depths over first year ice with snow density kept constant. Because the areal fraction and spatial distribution of FYI changes from year to year, this modification introduces a small degree of interannual variability into the contribution of snow data to sea ice thickness. We investigate this in Sect. (4.1.1).

## 2.5 Ice Type Data

Sea ice type data is required to modify W99 and create mW99. One popular product for this (e.g. Tilling et al., 2018; Hendricks and Ricker, 2019) is an operational product from the EUMETSAT Ocean and Sea Ice Satellite Application Facility (OSI SAF, www.osi-saf.org). However, this data series begins in March 2005. This is after our study begins (in October 2002).

A similar product exists, published by the Copernicus Climate Data Store (CDS, www.cds.climate.copernicus.eu; Aaboe, 2020). This product's underlying algorithm is adopted from the OSI SAF processing chain, but has been modified to produce a consistent record compatible with reanalysis. Furthermore, the CDS product only assimilates information from 'passive' satellite radiometers, whereas the OSI SAF operational product assimilates additional data from 'active' scatterometers. Despite these differences, a brief comparison of the products reveals a high degree of similarity.

It would be possible to use the CDS product prior to the beginning of the OSISAF product in 2005, but this approach raises issues surrounding the continuity of the products across the 2005 transition. Since our investigation focusses on trends and variability, we prioritise a consistent record and opt to use the CDS ice type product for the entirety of our study.

Both ice type products occasionally include pixels of ambiguously classified ice. We implemented a very simple interpolation strategy to classify these points while creating our mW99 data, although they are rarely present in winter within the regions analysed in this paper. Where the ambiguous pixels are generally surrounded by a given ice type then they are classified as the surrounding type. In the case where the ambiguous classification is on the boundary between the two types, the snow depth was not divided by two.

## 2.6 SnowModel-LG

To investigate the impact of variability and trends in snow cover on regional sea ice thickness we use the results of SnowModel-LG (Liston et al., 2020; Stroeve et al., 2020). SnowModel-LG is a Lagrangian model for snow accumulation over sea ice; the model is capable of assimilating meteorological data from different atmospheric reanalyses (see below) and combines them with sea ice motion vectors to generate pan-Arctic snow depth and density distributions. The sea ice motion vectors used were from the Polar Pathfinder dataset at weekly time resolution (Tschudi et al., 2020). SnowModel-LG exhibits more significant interannual variability than mW99 in its output because it reflects year to year variations in weather and sea ice dynamics.

SnowModel-LG includes a relatively advanced degree of physics in its modelling of winter snow accumulation. The model creates and merges layers based on precipitation and snowpack metamorphism. The effects of sublimation, depth-hoar formation and wind-packing are included. However, the effects of snow loss to leads by wind and extra snow accumulation due to sea ice roughness are not included. Furthermore, the heat flux to the snow is not sensitive to the thickness of the underlying sea ice.

SnowModel-LG creates a snow distribution based on reanalysis data, and the accuracy of this snow data is unlikely to exceed the accuracy of the input. There is significant spread in the representation of the actual distribution of relevant meteorological parameters by atmospheric reanalyses (Boisvert et al., 2018; Barrett et al., 2020). The results of SnowModel-LG therefore depend on the reanalysis data set used. However, the data product used has been tuned to match Operation Ice Bridge derived snow depths during spring time, and snow depth differences between the reanalysis products were found to be less than 5 cm (Stroeve et al., 2020). We note that the vast majority of Operation Ice Bridge flights were over the Beaufort Sea and the Greenlandic side of the Central Arctic, which is generally covered by multiyear ice. It is therefore conceivable that the scaling factors would be different if FYI were better sampled by OIB. The time-averaged regional differences between SnowModel-LG runs forced by ERA5 and MERRA2 reanalysis data are shown in Fig. (S3). The SnowModel-LG data used in this study are generated from the average of SnowModel-LG runs forced by the two reanalysis products. The SnowModel-LG data is provided on the same 25 km EASE grid as the ESA-CCI radar freeboards described above at daily time resolution. We averaged this daily product to produce monthly gridded data for combination with the monthly radar freeboard data.

## 2.7 NASA Eulerian Snow on Sea Ice Model (NESOSIM)

To support and broaden the impact of our findings, we repeat our analyses with NESOSIM data from 2002-2015 (Petty et al., 2018b). NESOSIM data is released on a $100 \times 100$ km grid which was interpolated to the $25 \times 25$km EASE grid of the SnowModel-LG and radar freeboard data. NESOSIM runs in a Eulerian framework and like SnowModel-LG can assimilate precipitation data from a variety of reanalyses data. In contrast with SnowModel-LG's multi-layered scheme, NESOSIM uses a two-layer snow scheme to represent depth-hoar and wind-packed layers. To define these layers, it assimilates surface winds and temperature profiles from reanalysis. Wind-blown snow loss is parameterised to leads using daily sea ice concentration fields (Comiso, 2000, updated 2017).

In this study we use data from a NESOSIM run initialised on the 15th August for each year. The initial depth was produced by a 'near-surface air-temperature-based scaling of the August W99 snow depth climatology'. This is a linear scaling based on the duration of the preceeding summer melt season. Snow density was initialised using the August snow-line observations of Soviet NP drifting stations 25, 26, 30 and 31. Data from the most recent publicly available drifting stations were chosen to maximise their relevance in a changing climate.

## 3    Methods

### 3.1    Contributions to thickness determination from snow and radar freeboard data

We now identify that the height correction due to slower radar pulse propagation in snow scales in almost direct proportion to the total mass of penetrated snow ($m_s$; Fig. S1). As such, it can be easily combined with the change to the floe's hydrostatic equilibrium from snow loading (also linearly dependent on $m_s$) to make one transformation to modify Eq. (2) such that:

$$SIT = f_r \frac{\rho_w}{\rho_w - \rho_i} + m_s \frac{\rho_w}{\rho_w - \rho_i} \times 1.81 \times 10^{-3} \tag{3}$$

Physically, the first term of Eq. (3) corresponds to the SIT were the sea ice known to have no snow cover. The second term is the additional sea ice thickness that is inferred from knowledge of the overlying snow cover. SIT has been decomposed into linearly independent contributions from radar-freeboard data and snow data. This allows the contributions of the two data components to SIT to be assessed independently. A derivation of the $1.81 \times 10^{-3}$ coefficient is available in the supplementary material.

We highlight here that our expression in Eq. (3) of the contribution of snow data to SIT determination solely in terms of snow mass is technically convenient for using W99 to estimate sea ice thickness, as quadratic fits of density (unlike depth and snow water equivalent) are not publicly available for all months. This has led to the required density values often being set to a constant value or 'backed out' by dividing the published SWE distributions by the depth distributions.

Eq. (3) and its factor of $1.81 \times 10^{-3}$ allow the simple expression of the theoretical change to the radar freeboard under rapid snow accumulation or removal. Making $f_r$ the subject of the equation and assuming SIT constant we find:

$$\frac{\partial f_r}{\partial m_s} = -1.81 \times 10^{-3} \quad (m/kgm^{-2}) \tag{4}$$

We stress that the above equation assumes total radar penetration of overlying snow, an assumption discussed in Sect. (5.3.1). As well as allowing independent analysis of the radar and snow data contributions to SIT at a point, the linearly independent nature of Eq. (3) in terms of $f_r$ and $m_s$ allows for a simple calculation of the average SIT in a region ($\overline{SIT}$) as:

$$\overline{SIT} = \overline{RF} + \overline{Snow} \tag{5}$$

Where $\overline{RF}$ and $\overline{Snow}$ represent the spatial averages of the first and second terms of Eq. (3).

### 3.2    Assessing Snow Trends and Variability at a point

In Sect. (4.1) we briefly compare the statistics for trends and variability at drifting stations published in Warren et al. (1999) with those introduced by mW99 and SnowModel-LG at a given point. We carry out this analysis to establish that the mW99 variability and trends at a given point (chosen as pixels on a 25x25 km EASE grid) are considerably smaller than those observed at drifting stations.

The monthly interannual variability (IAV) values published in Warren et al. (1999) are calculated as the standard deviation of the snow depths at drifting stations when compared to the climatology at the position of the stations. The IAV values at a point-like drifting station in a region will therefore naturally be higher than the IAV of the region's spatial-mean. As such, to compare IAV values from point-like drifting stations to mW99, we calculate the IAV at individual ice-covered points on a 25×25 km equal-area grid (Brodzik et al., 2012). These are all positive values, which we then average for comparison with the drifting stations. By regionally averaging the IAV values of many points rather than calculating the IAV of regional averages, we replicate the statistics of the point-like drifting stations.

However, the main part of this paper does not focus on trends and variability at a point (as measured by drifting stations), but instead investigates trends and variability in $\overline{Snow}$ and $\overline{SIT}$ at the regional scale (Sections 4.2 & 4.3). This variability is significantly lower than the typical variability at a point, as many local anomalies from climatology within a region are averaged out in the calculation of single, area-mean values which form a timeseries for each region.

### 3.3 Assessing Regional Interannual Variability

Sect. (4.2) of this paper focuses on the interannual variability in regional $\overline{SIT}$ which (treating $\overline{RF}$ and $\overline{Snow}$ as random, dependent variables) can be expressed thus:

$$\sigma^2_{\overline{SIT}} = \sigma^2_{\overline{RF}} + \sigma^2_{\overline{Snow}} + 2\,\mathrm{Cov}(\overline{RF}, \overline{Snow}) \tag{6}$$

Where the final term represents the covariance between spatially averaged radar freeboard and snow contributions. This covariance term can be expressed as $2r \times \sigma_{\overline{Snow}} \times \sigma_{\overline{RF}}$, where $r$ is the dimensionless correlation-coefficient between the variables and ranges between -1 and 1. To further explain this term, if years of high $\overline{RF}$ are correlated with high $\overline{Snow}$, then the covariance term will be high and interannual variability in $\overline{SIT}$ will be amplified. If mean snow depths are anti-correlated with mean radar freeboard across the years, interannual variability in $\overline{SIT}$ will be reduced.

$\overline{SIT}$, $\overline{RF}$ & $\overline{Snow}$ were calculated where any valid grid points existed on the 25x25 km EASE grid. Because of this, no average values were computed in the Kara Sea in October 2009 or 2012. Furthermore, no October values were generally available in the Barents Sea after 2008 (with the exception of 2011 and 2014). The impact of this on our resulting analysis is clearly visible in the top left panel of Fig. (10). We do not exclude the Barents Sea in October from our analysis because of the low number of valid points, but we do highlight the undersampling issue here. We continue to consider it because we do not find statistically significant declining trends with the data we have, so essentially we are reporting a null result. Our calculations of interannual variability in this month is inherently adjusted for the small sample size, but we nonetheless urge caution in interpretation of the values. The number of grid points available for averaging in each region in each month are shown in Fig. (S2).

The three terms on the right hand side of Eq. (6) correspond to the three unique terms of the covariance matrix of the two terms of Eq. (5). The main-diagonal elements of this 2×2 matrix correspond to the variance of the snow contribution and the radar freeboard contribution to sea ice thickness, terms one and two of Eq. (6). The off-diagonal elements are identical and sum to form the third term of Eq. (6).

We calculated this matrix for each region in each month to investigate the sources of regional interannual variability in $\overline{SIT}$ for the time period under consideration (2002-2018). The Central Arctic region is not sufficiently well observed by the Envisat radar altimeter (see Fig. 1), so the covariance matrix for the region was only calculated for the CS2 period (2010-2018).

In some cases a natural degree of covariance is introduced between the regional $\overline{Snow}$ and $\overline{RF}$ timeseries because they both display a decreasing trend. This 'false-variance' would not be present were the system in a steady state. As such, we detrended the regional timeseries prior to calculation of the covariance matrix. We found that doing this significantly decreased the value of the covariance term in Eq. (6) .

We consider the relative contributions of these three terms to $\sigma^2_{\overline{SIT}}$ in calculations involving mW99 and SnowModel-LG (Sect. 4.2). In light of these results, we then re-assess the statistical significance of regional trends in $\overline{SIT}$ using SnowModel-LG.

Detection of temporal trends in $\overline{SIT}$ is critically dependent on accurate characterisation of $\sigma^2_{\overline{SIT}}$. This is because conventional tests for trend exploit the known probability of a system with no trend generating the data at hand through variability alone (Chandler and Scott, 2011, p. 61). In this paper we argue that the $\sigma^2_{\overline{Snow}}$ term of Eq. (6) has been systematically underestimated through the use of a quasi-climatological snow data set (mW99). As an alternative to this we use the results of SnowModel-LG, a snow accumulation model that incorporates interannual changes in precipitation amount, freeze-up timing and sea ice distribution.

### 3.4 Assessing Regional Temporal Trends

In Sect (4.3) we examine temporal trends in regional $\overline{SIT}$ for each month of the growth-season (October - April), and decompose the results by sea ice type. It is stressed that these regional trends are each the trend of a single timeseries of spatially averaged thickness values, rather than the average of many trends in sea ice thickness at various pixels in a region. Regional trends were deemed statistically significant if they passed a two-tailed hypothesis test with p-value less than 0.05, with a null hypothesis of no trend. Trends were calculated for regional $\overline{SIT}$ over the Envisat-CS2 period (2002-2018) for all regions apart from the Central Arctic for which only CS2 data was available. We assess the relationship of these trends in $\overline{SIT}$ to trends in $\overline{RF}$ and $\overline{Snow}$ (Fig S19).

In Sect. (4.1.2) we show that basin-wide average snow depth and SWE is decreasing in SnowModel-LG in most months, but only in October for mW99. We point out here that (under the paradigm of total radar wave penetration of snow on sea ice) under-accounting for potential reductions in SWE may partially mask a decline in sea ice thickness, as reductions in radar freeboards are partially compensated by reductions in snow depths. From Eq. (5):

$$\frac{\partial(\overline{SIT})}{\partial t} = \frac{\partial(\overline{RF})}{\partial t} + \frac{\partial(\overline{Snow})}{\partial t} \tag{7}$$

## 4  Results

### 4.1  Comparison of point-trends and point-variability

#### 4.1.1  Low interannual variability in mW99 compared to drifting stations and SnowModel-LG

How does the variability in mW99 and SnowModel-LG at a given point compare to the values recorded at Soviet drifting stations published by Warren et al. (1999)? These values for interannual variability are not currently used in sea ice thickness retrievals (although they do contribute to uncertainty estimates in the ESA-CCI sea ice thickness product). Nonetheless, they offer a benchmark against which to evaluate the variability induced by mW99 at a given location.

Using the method described in Sect. 3.2 we find that the snow variability at a point from mW99 (Fig. 3, blue bars) is on average about 50% of the values recorded at the drifting stations (Fig. 3, green bars). By comparison, SnowModel-LG snow depth variability at a given point is significantly higher, ranging from ~75% of the drifting station values in October to ~115% by the end of winter.

We present this analysis of the point-like snow variability to illustrate that mW99 does not introduce enough variability at a given point to match that observed at drifting stations from year to year. Furthermore, the variability that does exist is confined to a distinct band of the Arctic Ocean (Fig. 4). This band represents areas where the sea ice type is not typically either FYI or MYI. Instead it is either switching between the two, or it is an area where FYI has replaced MYI during the period of analysis. In areas where sea ice type is temporally unchanging, snow variability is not present. This has implications at the regional scale as marginal seas with a consistent sea ice type experience unrealistically low $\sigma_{\overline{Snow}}$ in the mW99 scheme.

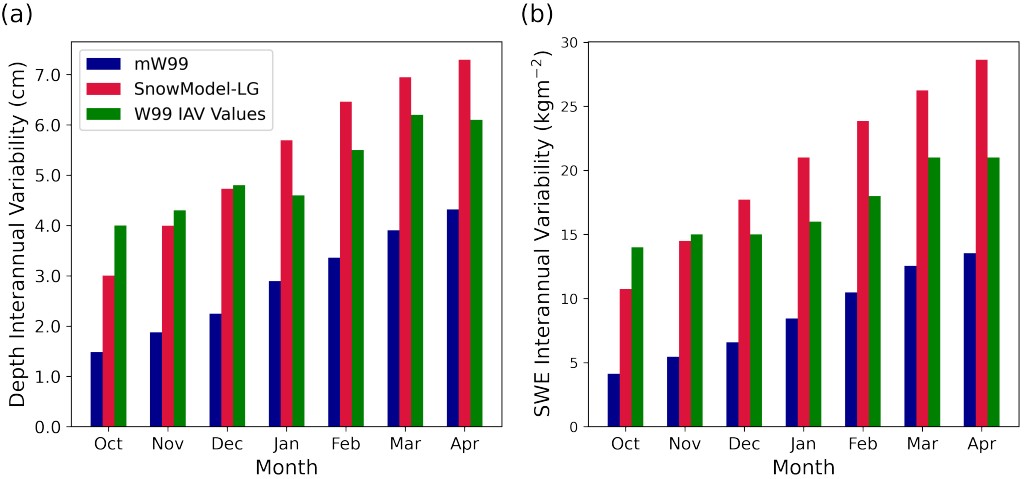

**Figure 3.** Interannual variability (2002-2018) in snow depth from mW99 and SnowModel-LG compared to the values given in Table 1 of Warren et al. (1999).

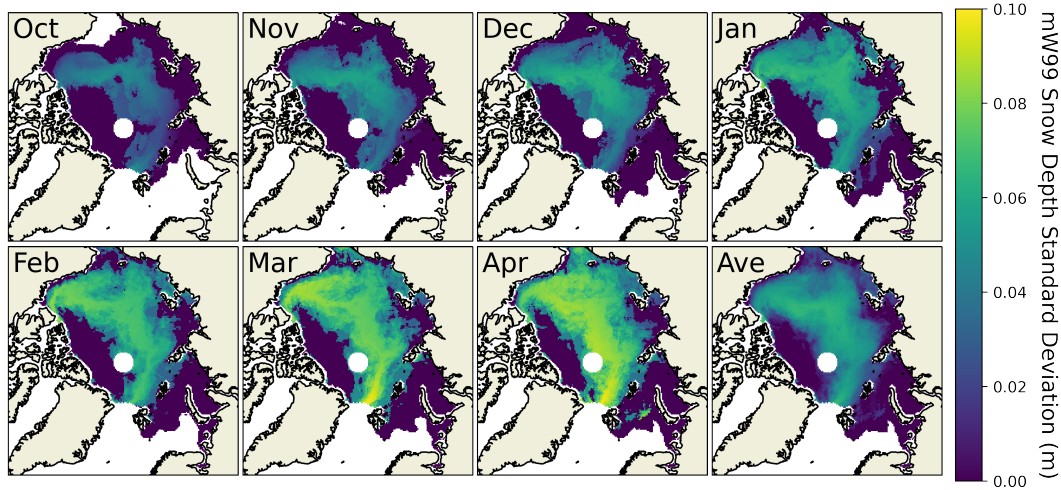

**Figure 4.** mW99 snow depth variability at each EASE grid point over the 2002-2018 period. This is calculated by generating a timeseries of snow depth at each point and then calculating the standard deviation of that timeseries. High variability is displayed in a band where sea ice type typically fluctuates from year to year. IAV is zero in areas that do not exhibit sea ice type variability, introducing unphysically low variability in SIT.

### 4.1.2 Lack of Temporal Trends in mW99 compared to SnowModel-LG and in-situ data

Weak trends exist at some points in the mW99 Arctic snow distribution due to the shifting distribution and abundance of first
year ice in the Arctic. In this section we briefly address their size, sign and veracity, leaving regional analysis until Sect. (4.2).

Values for SWE and depth trends measured by individual drifting stations are given in W99, but the values are not statistically
significant for any of the winter months, and as such are not displayed here. We instead compare the point-trends at all color
coded regions of Fig. 1 from mW99 and SnowModel-LG (Fig. 5).

We find that when we average the point-trends at a basin-wide scale, the only statistically significant trend (at the 5% level)
for mW99 snow depth is a positive one for the month of October (+0.11 cm/yr; Fig. 5). This increasing trend in snow depth is
in part due to the diminishing area of October FYI relative to that of MYI (Fig. S4), and in part due to the retreat of the October
sea ice into the Central Arctic where W99 exhibits higher snow depths and SWE. The increasing October areal dominance of
MYI is in part driven by delayed Arctic freeze-up (Markus et al., 2009; Stroeve et al., 2014). The area of sea ice over which
the W99 climatology is halved in October is therefore shrinking, and basin-wide mean snow depths in mW99 are increasing.
Trends in sea ice type fraction for each winter month are displayed in Fig. (S4), and monthly timeseries for mW99 SWE are
displayed in Fig. (S5).

Unlike mW99, SnowModel-LG exhibits statistically significant, negative point-trends for the later five of the seven winter
months (when averaged at a basin-wide scale). We identify two processes as responsible for this decreasing trend: the MYI
area is shrinking, so a smaller MYI sea ice area is present during during the high snowfall months of September and October
(Boisvert et al., 2018); also freeze-up commences later, so a lower FYI area is available in these months and more precipitation
falls directly into the ocean. Webster et al. (2014) observed a -0.29cm/yr trend in Western Arctic spring snow depths using
both airborne and *in situ* sources. This airborne contributions to this statistic included data over both sea ice types, and the
in-situ contributions included data from individual Soviet drifting stations from the Western Arctic. The statistic compares
well with the behaviour of SnowModel-LG (-0.27 cm/yr March; -0.31 cm/yr April), but is considerably beyond that of the
non-statistically significant trends of W99 and mW99.

What might the effects of this decline be on SIT at regional scales and larger? In terms of Eq. (7), models and observations
indicate that $\partial(\overline{Snow})/\partial t$ is negative on long timescales (Webster et al., 2014; Warren et al., 1999; Stroeve et al., 2020).
However, the use of mW99 effectively sets $\partial(\overline{Snow})/\partial t$ to zero, and to a positive value in October. This has the effect of
biasing $\partial(\overline{SIT})/\partial t$ high (and towards zero). Section 4.3 examines the effect of using SWE data with a more realistic decline
on regional $\overline{SIT}$ trends; this is mediated by the effects of higher interannual variability, which is examined in Sect. (4.2).

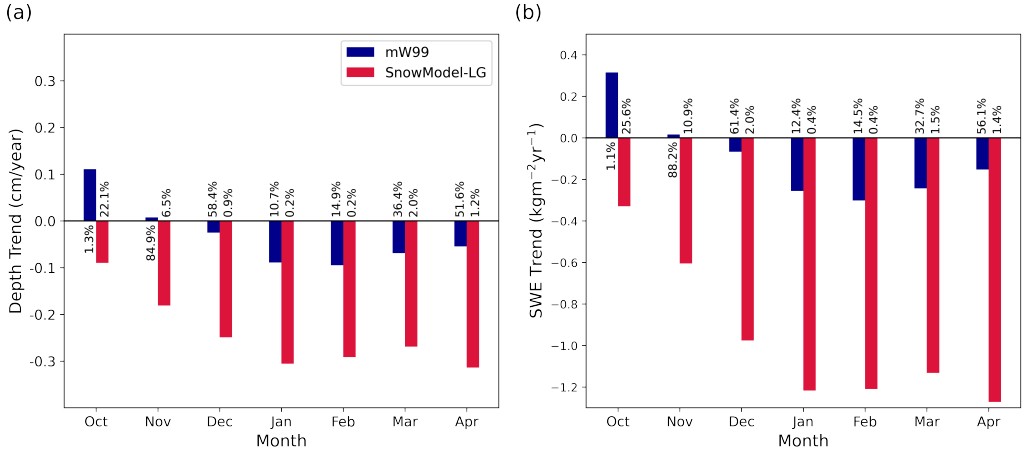

**Figure 5.** Basin-wide spatial average of point-like trends in (a) snow depth and (b) SWE, from mW99 and SnowModel-LG. Calculated for the Envisat-CS2 period (2002-2018). Significance values (in %) are given at the base of each bar. Only October trends for mW99 are significant at the 5% level, whereas significant negative trends exist in SnowModel-LG for December - April.

## 4.2 Realistic SWE Interannual Variability Enhances Regional SIT Interannual Variability

Having illustrated the deficiency of point-trends and point-variability in mW99, we now move on to the impact of snow data on SIT at the regional scale.

We calculate the interannual variability of detrended timeseries of the snow contribution to the thickness determination

($\overline{Snow}$) from mW99 and SnowModel-LG. We display some of these results in Fig. (6). We did this for every winter month (Oct-Apr) and for each region defined in Fig. (1). SnowModel-LG data produce more variable timeseries of $\overline{Snow}$ (i.e. higher values of $\sigma^2_{\overline{Snow}}$; c.f. Eq. 6). This is the case for all months, in all regions. For snow in the Kara Sea, mW99 introduces almost four times less interannual variability into $\overline{SIT}$ via $\overline{Snow}$ than SnowModel-LG in the April timeseries. This analysis is further broken down by sea ice type in Figs S7 and S8.

Having shown that SnowModel-LG's contribution to $\overline{SIT}$ is more variable than mW99, how does this increased variability propagate into sea ice thickness variability itself ($\sigma^2_{\overline{SIT}}$)? To answer this question, we must examine the way in which the snow contribution to SIT combines with data from satellite radar freeboard measurements. Having calculated the $\sigma^2_{\overline{Snow}}$ term of Eq. 6 (displayed in Fig. 6), we now turn to the $2\mathrm{Cov}(\overline{RF}, \overline{Snow})$ term. To assess this we calculate the magnitude and statistical significance of correlations between the detrended $\overline{RF}$ and $\overline{Snow}$ contributions to $\overline{SIT}$ in individual years, regions

and months.

To do this we calculated a monthly timeseries of $\overline{RF}$ and $\overline{Snow}$ for each region over the time-period 2002-2018 (with the Central Arctic being 2010-2018). Because we considered eight regions and seven months, this led to to 56 pairs of timeseries for $\overline{RF}$ and $\overline{Snow}$. We then detrended each of them. We then calculated the correlation between each of the pairs of detrended timeseries. We note here that the correlation between the timeseries is dependent on their relative position to a linear regression.

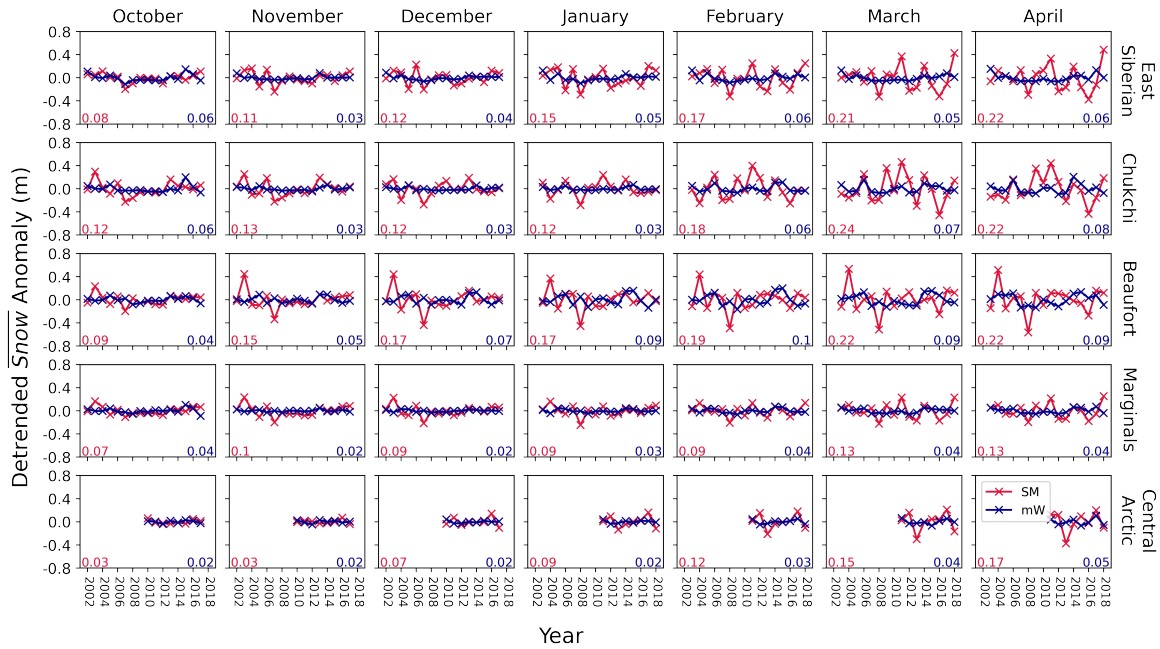

**Figure 6.** Detrended timeseries of spatially averaged snow contributions to sea ice thickness ($\overline{Snow}$) by region from W99 (blue) and SnowModel-LG (red). Standard deviation values are displayed for SnowModel-LG (lower left, red), and mW99 (lower right, blue). All regions are plotted in Supplementary Fig. (S6)

These correlation statistics are thus independent of the absolute magnitude of the values, their units, or any linear scaling of the axes. We therefore choose to present the correlations in Fig (7) without axes and scaled to the rectangular panels, so as to best show the relative positions of the points without extraneous numerical information.

We find statistically significant correlations between $\overline{Snow}$ and $\overline{RF}$ to generally range between 0.6 - 0.85 (Fig. 7). All statistically significant correlations were positive ones, and this was also the case when individual sea ice types were considered for each region. When all sea ice types were considered, the Laptev and East Siberian seas exhibited statistically significant correlations in five and six of the seven growth-season months respectively. The Barents Sea and the Beaufort Sea both exhibited one month of correlation, and the Central Arctic Region exhibited no months of correlation - the reasons for this are discussed in Sect. (5.4). When analysed as a single, large region, the 'Marginal Seas' area exhibits correlations in four of the seven months analysed, with the strength of these correlations increasing over the season.

We continued this analysis by breaking down the regions by sea ice type. The area of the Central Arctic sea ice covered with first year ice exhibits strong correlations (all above 0.8) in the later five months of the winter (Fig. S9).

When considering correlations over multi-year ice (MYI), the 'Marginal Seas' grouping exhibits correlations in the first four growth-season months (Fig. S10). The MYI fraction Central Arctic, Chukchi and Barents Seas exhibited no correlations. We note that this analysis is relatively sensitive to the detrending process. When performed without detrending, statistically

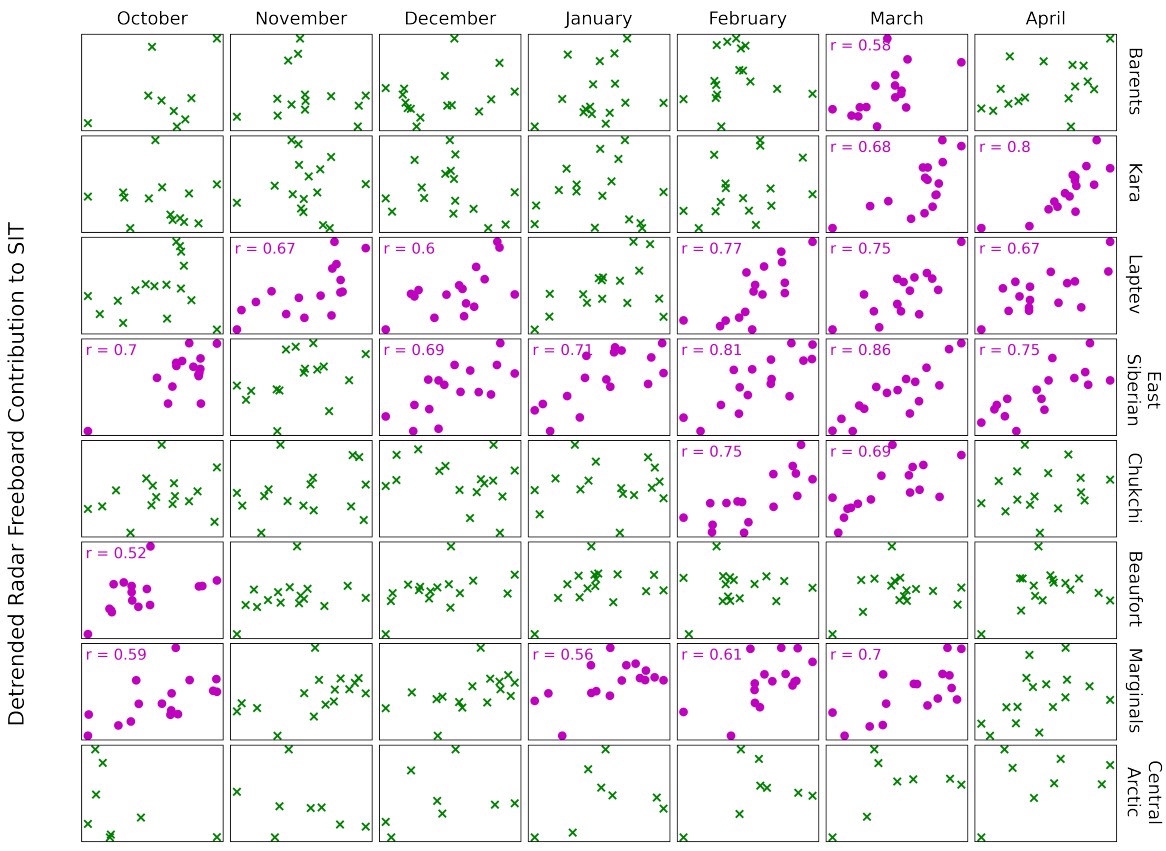

**Figure 7.** Covariability of contributions to sea ice thickness from radar freeboard and SnowModel-LG derived snow components over all sea ice types. Plots are colored with magenta when a a statistically significant correlation is present between the contributions (p>0.95). Analogous plots are displayed for the FYI and MYI components of the regions in Figs S9 & S10.

significant correlations are noticeably more common. This is because $\overline{Snow}$ and $\overline{RF}$ are both in decline in some areas, which introduces an inherent correlation from the trend.

Having identified and quantified regions and months of significant covariance between $\overline{Snow}$ and $\overline{RF}$ (Fig. 7), we are in a position to fully answer the question of how the increased variability of SnowModel-LG over mW99 (shown in Fig. 6) ultimately impacts $\sigma^2_{\overline{SIT}}$. We plot the three contributing components to $\sigma^2_{\overline{SIT}}$ for each region in each winter month (Fig. 8). We note that in the case of negative covariability between $\overline{Snow}$ and $\overline{RF}$, it is possible for $\sigma^2_{\overline{Snow}} + \sigma^2_{\overline{RF}}$ to be larger than $\sigma^2_{\overline{SIT}}$. This is not problematic because $\sigma^2_{\overline{Snow}} + \sigma^2_{\overline{RF}}$ does not represent a real quantity when the variables are not independent.

In the marginal seas $\sigma^2_{\overline{Snow}}$ overtakes $\sigma^2_{\overline{RF}}$ to become the main constituent of $\sigma^2_{\overline{SIT}}$ by end of the growth season (Fig. 8). This is particularly driven by the behaviour of the Beaufort and East Siberian Seas, where this relationship is clearly visible. In

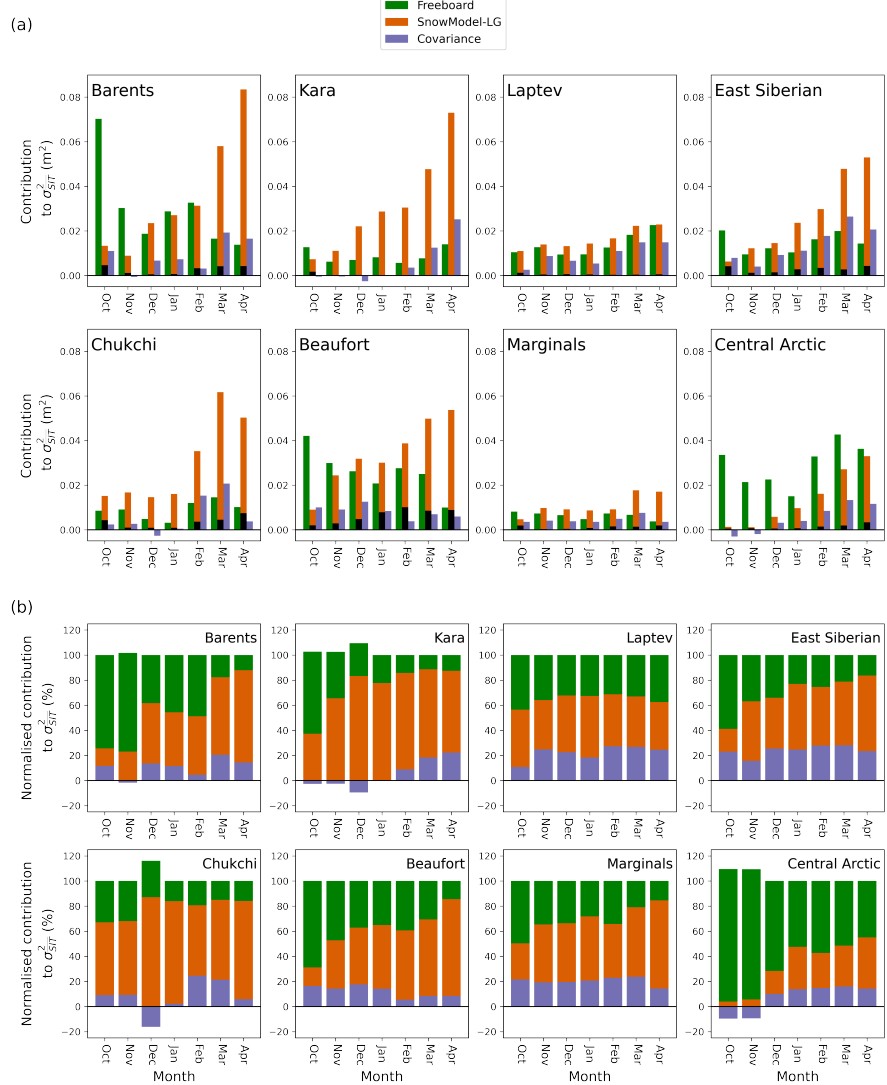

**Figure 8.** Constituent parts of $\sigma^2_{\overline{SIT}}$ of different regions. Bars represent the variance ($\sigma^2$) of $\overline{RF}$ and $\overline{Snow}$ and the covariance between the two. (a) illustrates the absolute variance contributions (b) illustrates their relative contributions. The variance of $\overline{Snow}$ in mW99 is indicated in panel (a) by a superimposed black bar. Snow contributes significantly more variability in the late winter than radar freeboard in most of the marginal seas.

the Central Arctic $\sigma^2_{\overline{RF}}$ narrowly remains the dominant component of $\sigma^2_{\overline{SIT}}$, throughout the cold season although $\sigma^2_{\overline{Snow}}$ plays an increasing role as the season progresses.

Covariance between $\overline{RF}$ and $\overline{Snow}$ makes relatively constant contributions to $\sigma^2_{\overline{SIT}}$ of the 'marginal seas' grouping in comparison to the other two components, but analysis of this grouping conceals more significant variation at the scale of the individual group members. The covariability term of Eq. (6) makes a larger contribution than radar freeboard variability itself at times, for example in the Kara and East Siberian seas at the end-of-winter, and for the Chukchi Sea in February and March. For the Central Arctic, the covariability term generally makes less of a contribution to total SIT variability than radar freeboard or snow variability individually, and is negative in the first two months of winter. We note that the covariability is almost always positive in the marginal seas with the exception of December in the Kara and Chuckchi Seas.

Finally, we directly compare the variability of $\overline{SIT}$ itself, when calculated using SnowModel-LG and mW99. We conduct this exercise in both absolute terms (Fig. 9a) and as a fraction of the regional mean thickness (Fig. 9b).

Calculation of regional SIT with SnowModel-LG reveals higher variability in all marginal seas of the Arctic basin in all months. When the marginal seas are analysed as a contiguous entity, the standard deviation is 0.09 m with mW99 and 0.16 m with SnowModel-LG. This represents an increase in $\overline{SIT}$ variability of 77%. For the Central Arctic this figure is considerably smaller, at 25%. When the individual marginal seas are considered, the largest increase was the Kara Sea (138%) and the smallest was the Beaufort Sea (35%).

One key aspect of interannual variability is how it compares to typical values. When IAV is expressed as a percentage of the regional mean thickness, the Barents Sea exhibits the largest increase when calculated with SnowModel-LG: the standard deviation (as a percentage of mean thickness) increases from 15% to 25%. When variability is viewed in this way, the increase in the Central Arctic is small (7.9% to 9.4%). Variability as a fraction of mean thickness is also highest in the Barents Sea when calculated with SnowModel-LG - whereas with mW99 this designation would go to the Beaufort Sea. When analysed as one area, variability (as a fraction of mean thickness) in the marginal seas transitions from being 7.5% of the mean thickness to 13.8% when calculated with SnowModel-LG.

We also note that MYI exhibits more thickness variability than FYI (both absolutely and relative to the sea ice type's mean thickness) in all the marginal seas (Fig. S11). For the marginal seas as a single group, MYI is roughly twice as variable in absolute terms. This is not the case in the Central Arctic, where the thickness variability of the individual sea ice types is highly similar (with FYI IAV slightly larger when calculated relative to regional mean thickness).

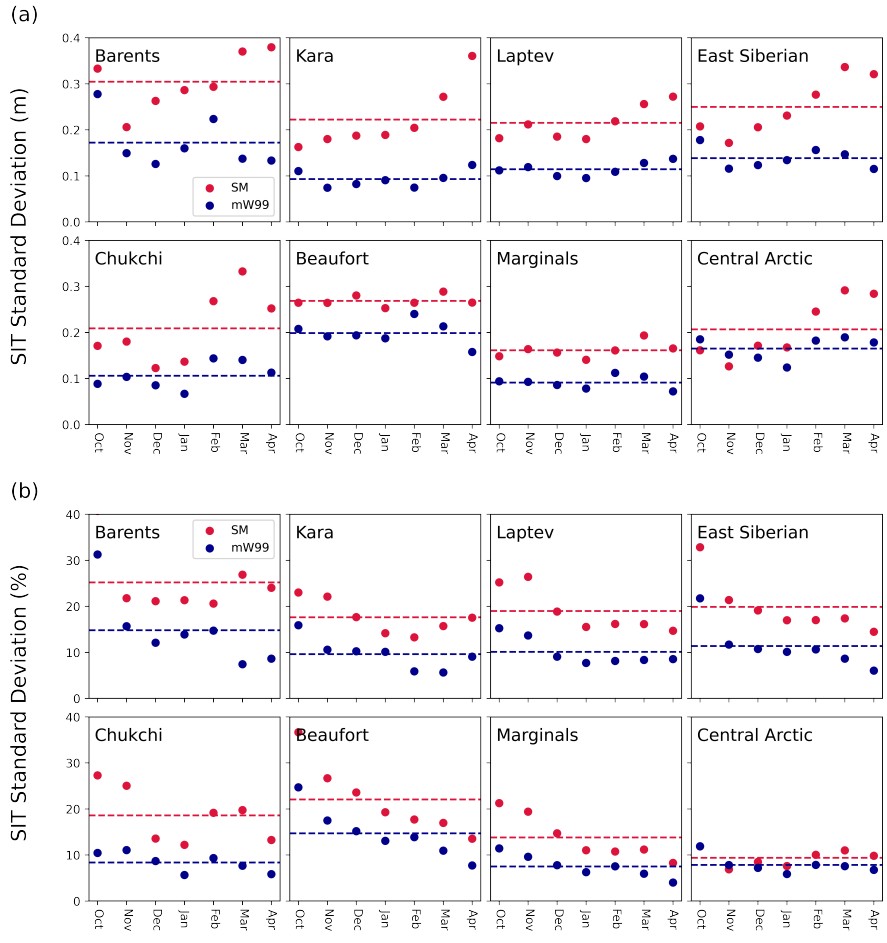

**Figure 9.** Standard deviation in sea ice thickness over the period 2002 - 2018 except for the Central Arctic: 2010-2018 (a) calculated in absolute terms (b) calculated as a percentage of the regional mean thickness over the period. Mean growth-season values shown with dashed lines. The individual detrended regional timeseries from which this figure is synthesised are available in Fig. (S12).

### 4.3 New and faster thickness declines in the marginal seas

As well as exhibiting higher interannual variability than mW99, SnowModel-LG $\overline{Snow}$ values decline over time in most regions due to decreasing SWE values year-on-year. Here we examine the aggregate contribution of a more variable but declining $\overline{Snow}$ timeseries in determining the magnitude and significance of trends in $\overline{SIT}$.

We first assess regions where $\overline{SIT}$ was already in statistically significant decline when calculated with mW99. This is the case for all months in the Laptev and Kara seas, and four of seven months in the Chukchi and Barents sea. The rate of decline in these regions grew significantly when calculated with SnowModel-LG data (Fig. 10; green panels). Relative to the decline-rate calculated with mW99, this represents average increases of 62% in the Laptev sea, 81% in the Kara Sea, and 102% in

the Barents Sea. The largest increase in an already statistically-significant decline was in the Chukchi Sea in April, where the decline-rate increased by a factor of 2.1. When analysed as an aggregated area and with mW99, the total 'Marginal Seas' area exhibits a statistically significant negative trend in November, December, January and April. The East Siberian Sea is the only region to have a month of decline when calculated with mW99 but not with SnowModel-LG.

We now turn our attention to new trends that stem from the use of SnowModel-LG over mW99 (Fig. 10; red panels). Our analysis reveals a new, statistically significant $\overline{SIT}$ decline in the Chukchi Sea in October (taking the number of months with a decline in SIT to five). Perhaps more significantly, the aggregated Marginal Seas region exhibits two new months of statistically significant declining $\overline{SIT}$ in October and February, taking the total number of declining months to six. No months in any marginal sea exhibited a statistically significant increasing trend in $\overline{SIT}$ (with either snow data set).

The Central Arctic region exhibits a statistically significant thickening October trend with both snow data sets (10 cm/yr and 9 cm/yr with SnowModel-LG and mW99). The region exhibits an additional month of increase in November when calculated with SnowModel-LG (7 cm/yr).

We also analyse these regional declines as a percentage of the regional mean sea ice thickness in the observational period (2002-2018). We observe the average growth-season thinning to increase from 21% per decade to 42% per decade in the Barents Sea, 39% to 56% per decade in the Kara Sea, and 24% to 40% per decade in the Laptev Sea when using SnowModel-LG instead of mW99. Five of the seven growth-season months in the Chukchi Sea exhibit a decline with SnowModel-LG of (on average) 44% per decade. This is much more than that of the four significant months observable with mW99 (25% per decade). We find the Marginal Seas (when considered as a contiguous, aggregated group) to be losing 30% of its mean thickness per decade in the six statistically significant months when SIT is calculated using SnowModel-LG (as opposed to mW99).

We further analyse these declining trends by sea ice type. This reveals the aggregate trends in the marginal seas to be broadly driven by thickness decline in FYI rather than MYI. We note that the FYI sea ice cover in the Kara and Laptev seas is in statistically significant decline with either snow product in all months. The FYI cover in the Barents Sea is also in decline for six of the seven winter months when calculated with SnowModel-LG. We find that (when analysed with SnowModel-LG) if any month in a specific marginal sea is in 'all types' decline, its first year ice is also statistically significantly declining.

## 4.4 Changes to the sea ice thickness distribution and seasonal growth

We now consider differences in the spatial sea ice thickness distribution introduced by a snow product with IAV. Because mW99 has low spatial variability in its SWE fields (the quadratic fits are relatively flat), it produces a more sharply peaked and narrow SIT distribution with lower probabilities of thinner or thicker sea ice in the months January - April. The SIT distribution also exhibits some degree of bimodality due to the halving scheme. This bimodality is to a large degree represented in the SnowModel-LG histograms - an encouraging result (Fig. S13).

The regional, seasonal growth rate is also similar when comparing calculations with SnowModel-LG and mW99 (Fig. S14). These rates were calculated over the period 2002-2018 with the exception of the Central Arctic which was restricted to the period 2010-2018. Among the most salient differences are the much smoother seasonal evolution of snow cover in the Barents Sea from SnowModel-LG and the decline in SWE from March to April in the Kara, Laptev and Beaufort seas with mW99

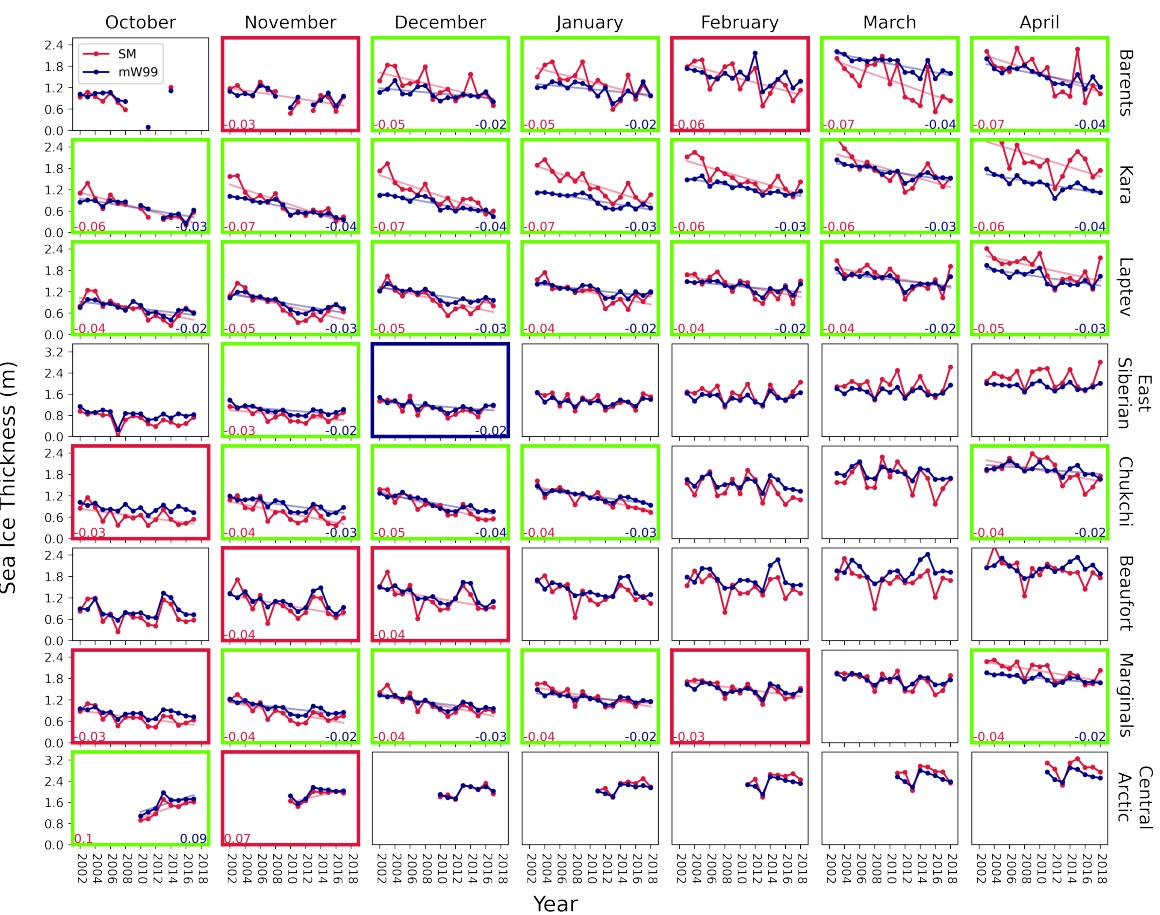

**Figure 10.** Regional $\overline{SIT}$ timeseries calculated using mW99 and SnowModel-LG. Note different y-axis scale for Central Arctic and East Siberian Sea. Panels featuring a statistically significant trend in sea ice thickness when calculated both mW99 & SnowModel-LG framed with green. Red frames indicate where trend is only significant when calculated with SnowModel-LG. Blue frames indicate where a statistically significant increase is detected with mW99, but not with SnowModel-LG. Where trends are statistically significant, trend lines are superimposed.

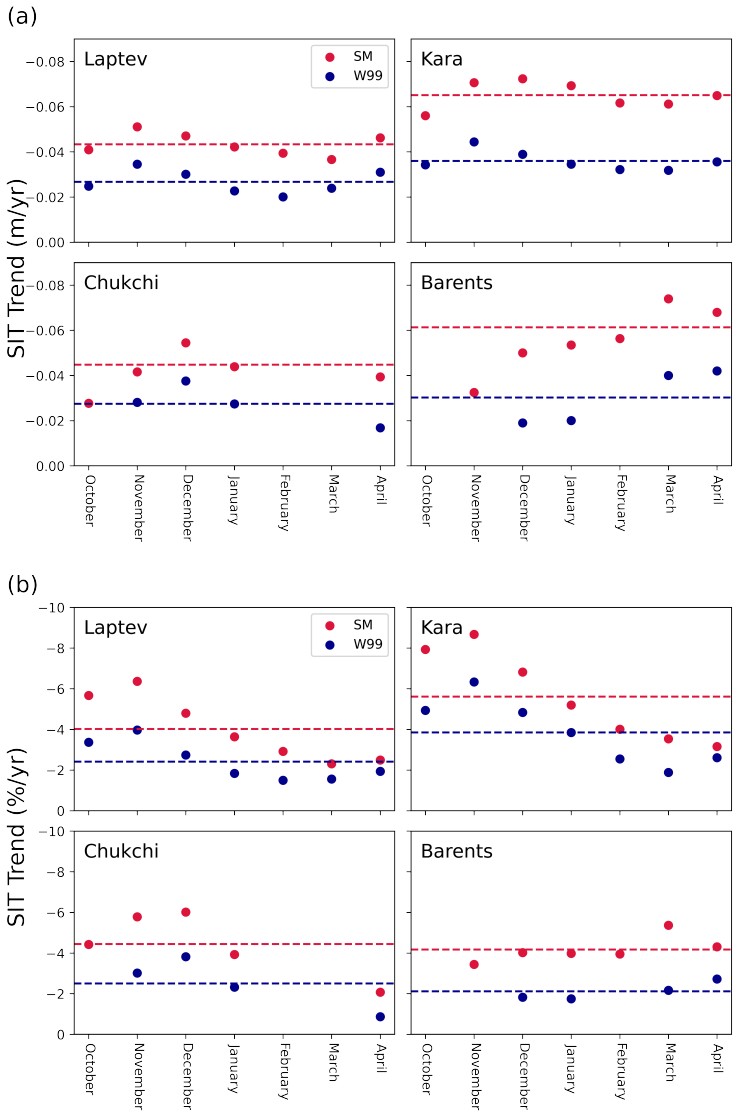

**Figure 11.** Sea ice thickness trends in the four marginal seas that exhibited robust trends in several winter months in the period 2002-2018. Average winter trend (calculated only from statistically significant months) from each snow product shown with dashed lines. Data points are only shown where a statistically significant trend is present for that month and for the relevant snow data.

(compared to a continued increase with SnowModel-LG). In the East Siberian and Laptev seas there is clearly a slightly lower seasonal growth rate when calculated with mW99, and this is also true to a lesser extent in the Chukchi Sea.

## 5   Discussion

### 5.1   Sensitivity of Findings to Choice of Snow Product

#### 5.1.1   Choice of Climatology - Combining AMSR2 with mW99

The most recent sea ice thickness product from the Alfred Wegener Institute (Hendricks and Ricker, 2019) makes use of a new snow climatology, generated by the merging of W99 with snow depth data derived from the AMSR2 passive microwave record. This is then applied with a halving scheme based on sea ice type in a similar way to mW99 (but with the AMSR2 component not halved). This likely improves the absolute accuracy of snow depths (and thus sea ice thickness), but does not resolve the issues discussed in this paper involving trends and variability. The modified AMSR2/W99 climatology functions in a very similar way to mW99 - a weak IAV is introduced in areas of interannually fluctuating sea ice type. Any trends will be the result of trends in the relative dominance of sea ice type. This was discussed in Sect. 4.1.2 and illustrated in Fig. S4: sea ice type trends are only significant in October and January, where they are weak.

#### 5.1.2   Choice of Reanalysis Forcing for SnowModel-LG

Barrett et al. (2020) reviewed precipitation data from various reanalysis products over the Arctic Ocean using records from the Soviet drifting stations, and found the magnitude of interannual variability to be similar. They further broke these data down to the regional scale using the same regional definitions in this paper, and found that this similarity persisted. Boisvert et al. (2018) conducted a similar analysis with drifting ice mass balance buoys, and found the interannual variability of the data sets to also be similar (although the authors found larger discrepancies in magnitude). These differences in magnitude however cannot be physical (as there is only one Arctic), and Cabaj et al. (2020) were able to bring precipitation estimates into better alignment using CloudSat data with a scaling approach. However this scaling approach preserved the interannual variability of the data sets, which Barrett et al. (2020) and Boisvert et al. (2018) found to be in comparatively good agreement. To investigate how this variability propagates into $\overline{Snow}$ variability, we calculate $\overline{Snow}$ timeseries from SnowModel-LG runs forced by both MERRA-2 and ERA-5 data and find their variability to be very similar (Fig. S15).

With regard to trends, we find that the two different reanalysis forcings generally introduce minimal differences in the SIT trends (Fig. S16). We do however find that small differences in SWE cause the $\overline{Snow}$ contribution of the MERRA-2 SnowModel-LG run to exhibit statistically signficant decline in regions and months where the ERA-5 run does not (with only a small change to the p-value). Analysis of the absolute $\overline{Snow}$ timeseries reveals them to be otherwise similar (Fig. S17).

We take these clear similarities as evidence that our findings are in principle robust to the choice of atmospheric reanalysis.

#### 5.1.3   Choice of Model - Comparison with NESOSIM

Some uncertainty is introduced into the spatial distribution of snow in a given year by SnowModel-LG snow parameterisations and simplifications, such as the lack of snow loss to leads. We therefore repeat our analyses with 2002-2015 data from the NASA Eulerian Snow On Sea Ice Model (NESOSIM;  Petty et al., 2018b).

We find that doing this increases the relative importance of snow variability to sea ice thickness variability (Fig. S18). We also observe that the NESOSIM calculations are considerably more similar to those done with SnowModel-LG than with mW99. NESOSIM replicates the increasingly dominant $\sigma^2_{\overline{Snow}}$ contribution to $\sigma^2_{\overline{SIT}}$ over the winter in the Marginal seas, and also replicates the higher contribution of $\sigma^2_{\overline{RF}}$ in the Central Arctic compared to both the individual and aggregated marginal seas. Striking resemblances are seen for the Kara Sea and the East Siberian Sea. Furthermore, the negative covariances for November in the Barents Sea and December in the Chukchi are replicated (albeit with significantly greater magnitude in the Barents Sea). NESOSIM also replicates the negative covariances in October and November in the Central Arctic, but also introduces negative covariance in December (unlike SnowModel-LG).

Because the NESOSIM data is only publicly available from 2002-2015, any underlying trends in the SIT timeseries are more challenging to detect because of the shorter observational period (by comparison to regions where all relevant data is available from 2002-2018). On the other hand, the calculated interannual variability is not reduced by the shorter timeseries, further obscuring any potential underlying trends. But despite these differences, both snow data sets produce statistically significant decline in all months in the Laptev Sea. NESOSIM reproduces six of the seven months of decline in the Kara Sea shown by SnowModel-LG, and three of the five in the Marginal Seas.

Further inspection of the individual data points across all regions and months reveals good agreement in regional SIT when calculated with either SnowModel-LG or NESOSIM - we take this as evidence that our findings concerning trends and variability over the longer 2002-2018 period are robust to the choice of reanalysis-accumulation model.

## 5.2 Study Limitations

### 5.2.1 Statistical Treatment

We have assumed in calculating single figures for variances that the interannual variability of the systems at hand is time-stationary. It is unclear whether this is the case, as the timeseries are limited in length and time-resolution and thus offer limited scope to test for stationarity. Furthermore we only tested for linear trends, when trends may in fact be non-linear. However, a visual inspection of Fig. (10) implies that this approximation is adequate on a qualitative level. Our trend tests also were two-tailed, with the null hypothesis that there was no trend. We could have formulated an alternate test where our null hypothesis was that the trend was positive. This would have given a higher number of statistically significant instances of negative trends, but we deemed this inappropriate as one of the regions (the Central Arctic) does exhibit significant positive trends with the two-tailed test.

## 5.3 Inter-Mission Bias between Envisat and CryoSat-2

An extensive validation exercise for the merged products indicated that although Envisat radar freeboards match well with CS2 freeboards in the Arctic overall, some biases do exist over specific ice types (ESA, 2018). In particular, analysis of the inter-mission overlap period indicates that Envisat freeboards were biased low (relative to CS2) in areas dominated by MYI, and high in areas dominated by FYI.

We first make the point that this will have a relatively minimal effect on our findings regarding interannual variability, as $\overline{Snow}$ is unaffected by this and $\sigma^2_{\overline{RF}}$ is likely relatively independent of the absolute magnitude of $\overline{RF}$.

With regard to trends, if Envisat radar freeboards (and thus $\overline{RF}$) are in fact biased high over FYI between 2002-2010 (relative to CS2), then the total trend in many regions dominated by FYI could potentially be smaller than calculated in this manuscript.

We do however add that our findings regarding the impact of declining $\overline{Snow}$ is unaffected by any inter-mission bias in $\overline{RF}$. Because the trend in $\overline{SIT}$ is determined by both $\overline{Snow}$ & $\overline{RF}$, the trend in $\overline{SIT}$ will always be more negative when calculated with downward trending data for $\overline{Snow}$.

### 5.3.1 The Effects of Incomplete Radar Penetration of the Snowpack

This investigation has been carried out within the paradigm of total Ku-band radar wave penetration of the snow cover (as suggested by Beaven et al. (1995)), however some in situ investigations have cast doubt on this. The issue was highlighted in an Antarctic context by Giles et al. (2008b) for ERS radar freeboards, and it was shown subsequently that significant morphological features in the snowpack (e.g. depth hoar, wet snow or crusts) enhanced radar scattering from within the snowpack (Willatt et al., 2010). For the Arctic, Willatt et al. (2011) found that airborne Ku-band radar backscatter in the Bay of Bothnia was returned from nearer the snow-ice than snow-air interface in only 25% of cases when the temperature was close to freezing, the figure increasing to 80% at lower temperatures. Nandan et al. (2017) observed that the presence of brine in the base of the snowpack can raise the scattering horizon by several centimeters. However, these investigations were often (but not exclusively) carried out at the end of the winter season or in the Sub-Arctic, when warmer temperatures may have increased the snow's brine volume fraction and diurnal forcing can drive rapid snow metamorphism. Both of these factors will be less prevalent in the colder months of winter. This analysis is therefore carried out using the imperfect historical assumption present in publicly available sea ice products (that of total penetration).

What would the effects of incomplete penetration of the snowpack be on our findings? As the height of the primary radar scattering horizon rises through the snow, the altimeter operation transitions from that of a radar altimeter to that of a lidar altimeter. Knowledge of overlying snow contributes positively to the inference of SIT in the case of a radar altimeter (i.e. the coefficient of $m_s$ term of Eq. 3 is positive). However, the influence of overlying snow on lidar-based SIT estimates is negative (i.e. the presence of more snow for a given measured radar freeboard implies less underlying sea ice). As the scattering horizon rises through the snowpack, the SIT contribution of snow therefore decreases, reaches zero (in the top half of the snowpack, the exact location depending on snow density) and proceeds to negative values. The result of potential incomplete penetration for our study is that the magnitude of the reported trend and variance underestimations is diminished. Were our investigation based on a similarly long timeseries of lidar freeboards combined with a snow climatology, one of our conclusions would be that diminishing snow cover is leading to *overestimation* of rates of decline in the marginal seas.

We finally note the potentially confounding influence of negative freeboard in regions such as the Atlantic sector of the Central Arctic region and the Barents Sea. In the case of high snowfall and low sea ice thickness, the sea ice surface can be depressed to the waterline or below. Beyond this point Eq. (5) no longer functions. The prevalence of negative freeboards has been studied by Rösel et al. (2018) and Merkouriadi et al. (2020), but has yet to be incorporated into any radar-altimetry based

sea ice thickness retrievals. This situation can be driven by storm tracks entering the Arctic from the Atlantic (but also the Bering Strait). These intrusions of warm air can also drive snow grain metamorphism, which may well affect radar penetration through the snowpack.

## 5.4 The Impact of Enhanced Variability from SnowModel-LG

When used instead of mW99, SnowModel-LG data increases the interannual variability of $\overline{SIT}$ in the marginal seas by more than 50%. The main way that this occurs is though increasing $\sigma^2_{\overline{Snow}}$ values (Fig. 6). The second and less significant way that $\sigma^2_{\overline{SIT}}$ is increased is through some positive correlations between $\overline{Snow}$ and $\overline{RF}$ values for individual months in some regions (Fig. 7). Because the two timeseries are positively correlated in some cases, $\sigma^2_{\overline{SIT}}$ is increased; for the Marginal Seas region

this covariance term makes up around 15-20% of $\sigma^2_{\overline{SIT}}$ (Fig. 8).

While values for interannual variability are given in W99, it was previously impossible to apply those values to either a given year or to fulfil Eq. (6). SnowModel-LG offers similar variability to the SWE statistics given in W99 (Fig. 3), and can generate a yearly timeseries of values. Furthermore it can be combined with radar freeboard data to generate all terms of Eq. (6) for a direct calculation of $\sigma^2_{\overline{SIT}}$.

Comparing our IAV values to the literature is challenging due to differences in the area over which other authors have calculated IAV values. Haas (2004) investigated the interannual variability of an area within the Transpolar Drift in the Central Arctic and Northern Barents Sea, and found a 0.73 m standard deviation. This is considerably higher than the values determined in this study, although this data was collected by electromagnetic sounding in late summer over a ten year period that does not overlap with this analysis. Laxon et al. (2003) defined a 'region of coverage', which essentially consisted of the marginal seas

considered in this analysis with the addition of some areas of the Canadian Archipelago and the Greenland sea. The authors found a variability of 0.24 m using W99 in this region of coverage over an eight year timescale. Unlike Haas (2004), this value is lower than our findings using either mW99 or SnowModel-LG. Similar to Haas (2004), the time period is considerably shorter and the geographical area is not identical. Finally, Rothrock et al. (2008) found interannual variability in SIT to be 0.46 m over a twenty-five year period (1975-2000), using submarine records from a variety of Arctic regions. It is likely that the

values in these studies differ due to the unequal spatial extent over which the IAV was calculated; averaging over a larger area reduces the IAV due to the averaging out of local anomalies.

## 5.5 The Impact of New and Steeper Trends in Mean Sea Ice Thickness

The replacement of multiyear ice with first year ice has been documented to be reducing Arctic-mean SWE on sea ice in spring (Webster et al., 2014). However, progressively later freeze-ups in the Arctic are also likely driving a reduction in mean SWE in

the early cold-season. This is because sea ice covers a relatively smaller area in the high precipitation months of September and October. When the sea ice area then expands with the progression of the growth-season, the newer sea ice has not been exposed to this snowfall. This mechanism is not accounted for in mW99, and as such snow depths do not decrease at a statistically significant level in any month.

In this study we have assessed how these negative trends in $\overline{Snow}$ propagate through into trends in $\overline{SIT}$. In every area where a statistically significant decline in radar freeboards is observed, a statistically significant decline in SnowModel-LG SWE is also observed (Fig. S19). In addition to this, SnowModel-LG also exhibits $\overline{Snow}$ decline in other months in the Beaufort and Barents Sea. As such, reductions in $\overline{Snow}$ usually act in concert with observed reductions in $\overline{RF}$, amplifying decline in $\overline{SIT}$. This relationship is illustrated by the fact that several months in several regions do not exhibit either a statistically significant decline in $\overline{RF}$ or $\overline{Snow}$ (Fig. S19), but despite this they do exhibit decline in $\overline{SIT}$ (Fig. 10). We note here that this 'co-decline' in $\overline{Snow}$ and $\overline{RF}$ is separate to the covariability presented in Sect. 4.2 and Fig. 7, as that was calculated from detrended data.

Because SnowModel-LG data features a steeper decline in $\overline{Snow}$ than mW99, a steeper decline is observed in the $\overline{SIT}$ of several regions. However, SnowModel-LG $\overline{Snow}$ contribution to $\overline{SIT}$ also exhibits significantly more variability, which acts to reduce statistical significance of $\overline{SIT}$ trends. Despite this compensating effect, the statistical significance of trends in $\overline{SIT}$ were generally greater than those calculated using mW99. Furthermore, statistically significant trends emerged in new months and new regions.

Kwok and Rothrock (2009) analysed 42 years of submarine records and the five year ICESat record. However, it is challenging to draw comparison with our results, as trends were gleaned from submarine track crossings and by comparing the thickness difference between the period of submarine observation and that of ICESat observations. Difficulty in comparison is further compounded by differences in regional designation and the area of the submarine data release (which is generally confined to the Central Arctic region where the radar altimetry timeseries is at best limited to the CryoSat-2 era). This is also the case for the updated analysis of Kwok (2018), who seasonally adjusted mean thickness values to match crossover points in submarine tracks in time and space.

Our findings of enhanced interannual variability and steeper decline have implications for Arctic stakeholders and the deployment of human infrastructure. The marginal seas are heavily used for the shipping of goods along the Northern Sea Route in summer (Eguíluz et al., 2016) and provide the setting for potential extraction of natural resources (Petrick et al., 2017). Furthermore, the season during which vessels may traverse the Northern Sea Route is lengthening. Higher variability in sea ice thickness may pose a challenge to the planning of this seasonal travel, particularly with regard to the need for ice-strengthened escorts for conventional vessels (Melia et al., 2017; Cariou et al., 2019). The enhancement of declining trends where they exist is perhaps of benefit these industries.

## 5.6 The interannual relationship between radar freeboard and snow depth

We finally consider the physical mechanisms behind positive or non-significant correlations between $\overline{Snow}$ and $\overline{RF}$ displayed in Fig. (7). Assuming total radar penetration of the snow cover, as snow accumulates on sea ice it should lower the local radar freeboard by a distance on the order of half its accumulated height (Eq. 4). This lowering is a result of physical depression of the sea ice surface and an increase in the radar ranging due to slower radar wave propagation in snow (in approximately a 60:40 ratio). Over short time scales (days to weeks), this would result in a negative correlation between local snow depth and local radar freeboard. This corresponds to a negative covariability term in Eq. (5) and is represented by purple bars in Fig. (8). Negative values are generally not seen, with the exception of October and November in the Central Arctic, November

in the Barents Sea and December in the Chukchi and Kara seas. Furthermore, snow is a highly insulating material and its accumulation limits sea ice thermodynamic growth. This would also bring about a negative correlation between snow depth and radar freeboard, lagged over a period of weeks.

The lack of negative correlations between $\overline{RF}$ and $\overline{Snow}$ from year to year is likely indicative of the timescale of our analysis. If present, the negative correlation implied by Eq. (4) and the mechanisms above must only be present on shorter timescales (e.g. days). So what drives the positive correlations between $\overline{RF}$ and $\overline{Snow}$ where they exist? One driver over FYI is likely sea ice age. Sea ice formed at the beginning of the season has a longer time to (a) grow thicker, and (b) accumulate snow. Both variables are therefore likely controlled by regional freeze-up timing, explaining the correlation. The combined evolution of $\overline{Snow}$ and $\overline{RF}$ anomalies as a function of regional freeze-up timings is likely to be the subject of future study. The relationship between MYI radar freeboards and accumulated SWE may also form an avenue for further study.

## 6 Summary

In this paper we used a novel approximation for the slowing of radar waves in snow to decompose the conventional method for estimating sea ice thickness into two contributions: one originating from radar freeboard data (from satellite altimeters), the other from snow data of varying provenance.

This allowed a regional assessment of the conventional impact of snow on variability and trends in sea ice thickness. We then used a new snow data set (from SnowModel-LG) with a more realistic magnitude of interannual variability and trends to calculate the regional sea ice thickness timeseries.

We found that interannual variability in average sea ice thickness ($\sigma^2_{\overline{SIT}}$) of the marginal seas was increased by more than 50% by accounting for variability in the snow cover. On a seasonal timescale we find that variability in the snow cover makes an increasing contribution to the total variability of inferred sea ice thickness, increasing from around 20% in October to more than 70% in April.

We also observed that the trends in SnowModel-LG data propagated through to the SIT timeseries, amplifying decline in regions where it was already significant, and introducing significant decline where it did not previously exist. This occurred in spite of the compensating effect of enhanced interannual variability.

*Author contributions.* JCS, JCL, MT and RW proposed and conceptualised the study. VN and GEL provided extensive feedback on manuscript and GEL provided the SnowModel-LG data. RDCM carried out the main analysis. All authors contributed to the write-up.

*Code and data availability.* The code used for all analysis and visualisation was written in Python 3.6 and is available at github.com/robbiemallett/SnowModel-LG_SIT_Impacts. The radar freeboard data from Envisat and CryoSat-2 is available from the ESA CCI initiative at climate.esa.int/en/odp/#/project. The NESOSIM snow data is available from the NASA Cryospheric Sciences Laboratory

website at earth.gsfc.nasa.gov/cryo/data/nasa-eulerian-snow-sea-ice-model-nesosim. SnowModel-LG data is currently pending review at the US National Snow and Ice Data Center and will be available at doi.org/10.5067/27A0P5M6LZBI. Code and data last accessed 2020/2/20.

*Competing interests.* The authors declare no competing interests.

*Acknowledgements.* This work was funded primarily by the London Natural Environmental Research Council Doctoral Training Partnership grant (NE/L002485/1). JCL acknowledges support from the European Space Agency Living Planet Fellowship 'Arctic-SummIT' under grant ESA/4000125582/18/I-NS and the Natural Environmental Research Council Project 'Diatom-ARCTIC' under Grant NE/R012849/1. MT acknowledges support from the European Space Agency by project 'Polarice' under grant ESA/AO/1-9132/17/NL/MP, project 'CryoSat + Antarctica' under Grant ESA AO/1-9156/17/I-BG and project 'Polar + Snow' under Grant ESA AO/1-10061/19/I-EF. JCS and MT also
acknowledge support from the Natural Environment Research Council (grant no. NE/S002510/1). JCS and GEL acknowledge support from NASA grant 15-CRYO2015-0019 /NNX16AK85G. JCS acknowledges the support from the Canada 150 Chair Program. VN was supported by JCS, in part thanks to the Canada 150 Chair Program. RW was supported by NERC grant NE/S002510/1. JCL, JCS and MT were part funded by NERC Project 'PRE-MELT' under Grant NE/T000546/1.

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
