# Peer review of "Faster decline and higher variability in the sea ice thickness of the marginal Arctic seas when accounting for dynamic snow cover"

_The Cryosphere, 2020_

## Referee Comment (RC1) · Stefan Kern (Referee) · 23 Nov 2020

Review of

Faster decline and higher variability in the sea ice thickness of the marginal Arctic seas

by

Mallett, R. D. C., et al.

Summary: This very interesting paper illustrates the potential improvement in the credibility of trends in and inter-annual / intra-basin variations of Arctic sea-ice thickness estimates from satellite radar altimetry. This is achieved by a comprehensive inter-comparison of the contribution of snow on sea ice on the retrieval of sea-ice thickness

from radar freeboard when using the Warren et al. (1999) snow climatology on the one hand and a physical model for snow properties driven by atmospheric reanalyses' precipitation and other relevant meteorological parameters on the other hand. As expected, the inter-annual variability of the snow contributions based on the model data is considerably larger than the one based on the Warren et al. (1999) data. The paper further convincingly demonstrates that the more realistic inter-annual variation and trend in the snow parameters obtained with the model yields a new overall picture of the variability and spatio-temporal development of the Arctic Ocean sea-ice thickness.

The paper is generally well written and will have considerable impact on the scientific community. It would benefit from some re-organization (see GC1). It is furthermore quite light when it comes to descriptions of data and methodologies used (GC2). Currently, one would not be able to re-produce the work done. The inclusion of Kara and Barents Sea I find quite a hypothetical move based on the data availability and suggest to consider removing those from the analysis (GC3). Finally, there is a number of open points to discuss when it comes to the illustration and interpretation of the results presented.

In the following you will find my list of general comments (GC), specific comments and some suggestions to mitigate typos and editoral issues - all for the main manuscript - followed by a short list of things I found worth to consider in the supplementary material.

General comments:

Title: While your main conclusion supports the title in general, it is in some way misleading. The main focus of the paper is on the illustration that a snow depth climatology is not well suited to compute credible trends in sea-ice thickness estimates derived from satellite altimetry with such a snow depth as input for the freeboard to thickness conversion. In your paper, this is illustrated by usage of data from a numerical model which has experienced limited validation. Hence, albeit the improvement using these model data is obvious it is not necessarily the truth either. Hence, instead of formulating the

title as a fact I suggest to include points of the above-stated.

GC1: I strongly recommend to re-organize the paper. Most of the explanations / motivations given in the subsections 1.1 and 1.2 are tied relatively close to Section 3 and should be combined with that section. In addition, subsections 1.1 and 1.2 refer to data and regions denoted in Section 2. Hence: Remove 1.1 and 1.2 and put it into Section 3. Let Section 2 start right behind the "true" introduction. That way the data sets used in 1.1 and 1.2 would be introduced adequately beforehand which eases reading and which reduces the number of open questions.

GC2: Both, the description of the data used as well as of the methodologies used lack some clarity and/or do not contain all information required. One good example: The ESA-CCI radar freeboard data set used comprises data of two different satellites with some overlap. It is not clear from the description in the data how long the Envisat and how long the Cryosat-2 part of the data used is - plus a motivation of the choice made - plus a discussion about the biases between the radar freeboards of these two satellites, which have a different sign based upon the region. Some of the descriptions also appear to contain errors which ask for re-phrasing.

GC3: The overall credibility of the paper would benefit from a more critical consideration of the application area of the Warren et al. (1999) climatology. Sampling density, number of observations, as well as the distribution of the snow depth observations over time combined with the usage of a polynomial fit limits the usefulness of these observations in the regions Kara Sea and Barents Sea. One good solution would be to omit these regions.

Specific comments

Line 18: " ... it determines whether floes ridge or raft ... " –> My take on this would be that this happens at rather small ice thicknesses, i.e. around 20-30 cm. I am therefore not so sure whether this is such an important physical role of the sea-ice thickness.

Line 21: "... with thin ice favoring melt pond formation ..." Why is that? Because there is little snow, which melts away more quickly than on thick is with a thicker snow cover? Otherwise I don't see a pressing reason why melt pond formation, which is basically driven by downward short- and longwave radiation should occur more easily on thin than on thick sea ice.

Line 76: "below the waterline" –> Why this part of the total SIT is from below the waterline? I find this addition confusing because it implies that the sea ice is thicker when it is snow covered - which is not necessarily the case. Equation (2) refers to a SIT which is similar in both cases, bare or with snow cover. It could be 1 m, 2 m, whatsoever, with or without snow. The quantities that change are the sea-ice freeboard and the radar freeboard. It seems you want to express that in case of a snow cover the part of the sea ice that is below the waterline is larger than in case of bare ice. Usually this part below the waterline is called draft. It might hence make sense to re-phrase this sentence a bit to avoid confusion.

Line 83: "assumes total radar penetration of overlying snow" –> How about penetration of the Ku-Band signal into the sea ice? Given the different near-surface sea-ice salinity and densities of MYI compared to FYI one might need to also make a comment on this issue?

Lines 109/110: "as quadratic fits ... without corresponding fits of density." –> I don't agree. Warren et al. (1999), page 8, writes about 2-dimensional density fits. Yes, only a May map is shown but maps are derived for all months. And it is the SWE which is computed, not the other way round. Please rewrite this paragraph accordingly.

Lines 121-124: I suggest to provide information whether and to which extend this modification of W99 has been implemented by follow-on studies, e.g. Tilling et al. ? Kwok and Cunningham 2015 ...

Lines 126-127: "are not currently used in sea ice thickness retrievals" –> I am wondering whether these variabilities are input into the uncertainty estimates provided along

Interactive
comment

with the ESA-CCI SIT products? It might be worth to check.

Lines 130-133: "As such, ... values." –> Will this described in more detail in section 2? No it will not. Are positions of real drift stations used? If yes, which? If not: Isn't taking into account ALL ice covered grid cells of a 25 km grid providing a substantially different statistics - compared to the few drift stations used in the W99 climatology (their Fig. 1)? What is the time period considered? The description of this analysis step is lacking key details and should be re-written.

Figure 2: Fig. 2 could be connected more easily to Fig. 1 if you'd use snow depth instead of SWE. It is not entirely clear how these maps are derived. I note that the ice edge in the Kara / Barents Seas looks a bit weird for months DEC and JAN.

Line 136: "this results" –> Which? Not clear to what this refers.

Line 138: "where the ice type typically varies from year to year" –> How did you define this?

Lines 157-162: "We find ... (S3)." –> One question upfront: What is exactly the region you are considering here? The central Arctic Ocean? Laptev Sea included? Kara Sea? I am just asking because at a certain point in winter the entire region considered should be ice covered and the FYI fraction hence be only a function of the MYI extent.

I buy that there is a the decreasing fraction of FYI relative to the total ice extent in October due to later freeze-up. I don't agree, however, that this is the sole reason for your observation with respect to the trend mW99 snow depth and SWE. I believe an issue to consider is that the MYI coverage retreats more and more to those regions where W99 has maximum snow depth. Hence the relative fraction of MYI grid cells with comparably thick snow is increasing which to my opinion can result in a higher mean W99 snow depth (and SWE) for the MYI part of the sea ice in October.

Lines 164-166: "Several ... year to year. –> I don't find this formulation particularly clear. I find the "cannot accumulate snow from year to year" not to well chosen in the

light of mostly complete snow melt during summer - also on multiyear ice. I'd state that there are two reasons for the observations with SnowModel-LG: 1) The MYI area shrinks. Hence your sentence about "a lower [smaller] ice area is exposed to snowfall in September/October fits well. 2) Freeze-up commences later, hence new seasonal ice either has not yet formed or is too thin to carry / accumulate snow resulting in a substantial amount of the precipitation falling as snow being dumped into open water. In short also here your sentence about "a lower ice areas ..." applies, meaning you can, to my opinion, delete that extension "also the later ... year."

Line 167: "Webster et al. ..." –> I am wondering whether the "in situ sources" mentioned in the context of Webster et al. (2014) are i) also representing FYI and ii) aren't complemented with information from airborne operation ice bridge data [in which case these are not "in situ" anymore]. Please check! If their data indeed represent FYI and MYI then it might be worth to mention that explicitly in your manuscript.

Lines 187-189: "Where sea ice ..." –> Could you please comment on whether this second data set is similar to / consistent with the OSI-SAF one? What is the basis? Given the fact that you investigate quite short time series in your paper and put quite some weight on different ice types it is important that thes two data sets are consistent to each other, i.e. provide a seamless continuous spatial FYI/MYI fraction distribution without a jump in total regional FYI and MYI extent from, e.g., Feb 2005 to March 2005.

I note that you also use NSIDC ice-age data and one could ask the question: why didn't you use ice-age data throughout the entire study?

Section 2.3: Please comment on two issues. 1) The sampling on which W99 is based has large regional variations with substantial differences between marginal seas such as the Kara or Barents Seas compared to the central Arctic. How does the lack of reliability of W99 in these partly undersampled regions influence your results - particularly in the two regions mentioned above? 2) Snow depths / SWE in the Kara / Barents Sea do - for the same reason - depend a lot on the extrapolation / fit function used. How

does this influence your results? Aren't the snow depth values in these regions too hypothetical to be adequately used in your study? Wouldn't it make sense to exclude the Kara and Barents Seas? To my opinion it would make your study considerably more credible. And it would potentially safe some space.

Section 2.4: Isn't the EASE grid a polar aspect of the Lambert Azimuthal Equal area grid? I'd suspect no re-gridding is required. What is the grid resolution of the radar freeboard data? What is the time period (years, months of the year) for which these data are available and used by you? How did you treat the overlap of Envisat and Cryosat-2? Key information is lacking here. It might also be worthwhile to take a look into the validation report of the SIT / freeboard data set used (see e.g.: https://icdc.cen.uni-hamburg.de/fileadmin/user_upload/ESA_Sea-Ice-ECV_Phase2/SICCI_P2_PVIR-SIT_D4.1_Issue_1.1.pdf ). It provides some information about how "consistent" the two "merged" data sets are. Taking this information into account and discussing the potential biases (which still exist) I rate mandatory for a paper which so much relies on the analysis of this 17-year long time-series with a change of sensor right in the middle of the time series.

Line 206: "ice motion vectors" –> Which ice motion vectors? Please provide this information - including the temporal resolution and the version of that ice motion data set used - in your manuscript. In addition: "pan-Arctic snow depth and density distributions" –> Please provide a spatial and a temporal resolution as well as the domain. While the paper focuses a lot on snow depth I am wondering how snow densities obtained with SnowModel-LG compare to W99 ones?

Line 210: "snow-ice accumulation" –> Please explain what "snow-ice accumulation" is. Do you refer to snow-ice formation at the basal snow layer?

Lines 216/217: "snow depth differences ... than 5 cm" –> This is a quite global statement. Is this an Arctic mean value? Is this the mean difference in SnowModel-LS realizations just for the grid cells co-located with the OIB data? What is the standard

deviation of this difference? Do the OIB data used to tune SnowModel-LG represent FYI conditions adequately?

Section 2.6: For a better understanding it might make sense to explicitly state whether precipitation and/or snow fall are assimilated into NESOSIM as well. The snow pack initialisation, is this covering both snow depth and density? As W99 data are monthly values, is this initialisation only done monthly, or are monthly values interpolated to daily values with which the model is initialised henceforth? You explicitly mention depth-hoar and wind-packed layers in the context of NESOSIM. Does this imply that SnowModel-LG does not represent such features? If not, then I suggest to be more specific in the description of what SnowModel-LG can do and what not.

Section 3.2: Please provide more details. How many grid cells with valid SIT measurements are requied to compute a regional mean SIT value? How about regional means of radar freeboard and snow depth / SWE? Did you compute these as well? How many valid observations are required for the results broken down into ice types (see Figures S4 and S5)? Please provide a reference for the "Wald test".

Line 249: I suggest to stress here once more what "Snow_overbar" is, that it is not the snow depth but the snow-depth contribution to the SIT retrieved from altimetry

Line 259: "individual year, regions and months" –> Not clear what you did. You used detrended time-series of monthly, region-mean values of RF and snow and computed the correlation between these time series separately for every month and every region?

Lines 263/264: "The Barents Sea ... correlation." –> I suspect this observation is based on two completely different causes. For the Central Arctic the time series is just 9 years long. For the Barents Sea, neither is mW99 overly reliable nor are RF values overly reliable - especially during the Envisat period. Another, more general comment: The RF data for region Central Arctic are considerably more robust in terms of the number of valid observations contributing to the RF values used.

Figure 6: Not clear what is shown magnitude-wise on x- and y-axes. The same applies to Figures S6 and S7.

Line 286: "but analysis ... regions." –> Not clear what you mean here.

Line 287: "The covariability ... contribution" –> This discussion focuses on radar freeboard. It does not comment on the observation that at the beginning of winter (Oct.) the fraction of SIT IAV that is explained by RF-Snow covariance is larger than snow IAV.

Figure 7: Can you please check whether your representation of "Fraction of Total Variance (%)" is correct? I mean, ok, if the dimensionless factor rho is negative then the covariance term in Eq (5) gets a negative sign. Therefore you plot negative bars in panels (b). However, looking at the Central Arctic, November, this results in a fraction of radar freeboard IAV of about 110%, also the one for October is larger than 100%. I get a headache with this because a fraction cannot be negative (have you ever had a negative piece of cake?) and it can also not be larger than 100%, i.e. larger than the total (only if you order a medium size Pizza and get a large one instead). This applies then also to Figure S15 where the deviations from 100% are even larger. Again, I can see from Eq. 5 that it is mathematically correct. However, a positive covariability means that sigma_RF and sigma_snow are positively correlated while a negative one means that these quantities are anti-correlated. If we assume a very strong negative covariability of, say -90%, does that mean that the IAVs of RF and snow need to sum up to a fraction of the total of 190%?

Lines 327/328: "Perhaps more significantly, ..." –> This I don't find too convincing - also given the unknown uncertainty of these regional mean SIT values. I suggest to only mention these three new trends but do not hypothesize about the main reason.

Line 350: "... truncated SIT distribution ... thicker ice." –> This relatively global statement is not supported by Fig. S10 for all months. Particularly, I would not use the word "truncated". Truncated means that below or above a certain SIT values the area

[Figure]

occupied by these SIT bins is abruptly zero.

Lines 351/352: Please see my comment at Figure S10: You need to provide more details about how you derived this Figure. What is missing are binsizes and borders as well as the time-period for which the Figure is valid (2010-2018 I assume) as well as a statement here that this Figure is now showing a classical pdf but expresses the distribution in form of sea-ice area. In order to avoid confusion with the classical definition of sea-ice area which is sum of the area of ice covered grid cells weighed with the actual sea-ice concentration, you might want to rename your y-variable.

Line 353: "The regional, seasonal growth rate ..." –> What is the period considered?

Lines 372-374: I suggest to refer to the Boisvert et al. (2018) paper here (about the difference between Merra-2 and ERA-Interim) and in addition take into acount this paper: https://doi.org/10.1029/2019GL086426 . I am wondering whether Figure S12 is required. After all it confirms that your choice combining SnowModel-LG runs of ERA5 and Merra-2 is a good one.

Line 387: "replicates the higher contribution ..." –> Even though for the Central Arctic the data are just based on the period 2010-2015 = 6 years.

Linees 390/391: "underlying trends ... period" –> Ok ... but at the same time you use a shorter time period for the Central Arctic anyways. So this argument is not conclusive.

Lines 394-399: "As such ... 2018)." –> I suggest to shorten this part and perhaps either delete Figure 11 or move it as well to the Appendix. - which however, already contains a lot of additional material. But I feel that the paper would be still understandable with a few sentences highlighting the common findings between NESOSIM and SnowModel-LG

Line 427-428: "... these investigations ... warmer temperatures ..." –> I do not quite agree to this statement because, according to my knowledge, among the cases investigated and presented in Nandan et al. (2017) as well as in the later paper by Nandan et

al. (2020, https://doi.org/10.1109/JSTARS.2020.2966432 ) is a sufficiently high number of cases with cold snow; hence the observation of a rising scattering horizon is not uniquely tied to warmer temperatures.

Lines 434-440: "Knowledge ... depth." –> I can follow the physical reasoning here and also how it is connected with Equation (2). However, it might not be as straightforward as it is formulated, given the fact that equation (2) transforms into equation (4) in Kwok and Cunningham (2008, https://doi.org/10.1029/2008JC004753 ) for a lidar - hence introducing a different factor in front of the snow depth. Perhaps formulating these lines more like being your own hypothesis than being fact might be more appropriate?

Line 443: In the context of the paragraph ending here you could also comment on the impact of i) the increased likelihood of a flooded basal snow layer in regions of comparably thin sea ice / thick snow / high SWE as, e.g. the Barents Sea and part of the region Central Arctic facing the Atlantic and ii) of a potentially increased likelihood for enhanced snow metamorphism, like is typical for the Antarctic, even in the middle of winter due to intrusions of marine air-masses into the Arctic, e.g. via the Fram Strait.

Line 469: "in the high precipitation months ..." –> One could argue, though, that thanks to later freeze-up and the concomitant change in atmospheric moisture content (and perhaps also circulation) also shifts the maximum of the precipitation to a later time - and with that the seasonal sea ice would still be able to accumulate a fair amount of snow - particularly supported by the fact that a warmer atmosphere can hold more moisture and with that can lead to increased precipitation rates. This is hypothetical of course but among the possible scenarios.

Lines 504/505: "negative ... this is not seen." –> I thought the negative covariances between RF and Snow shown in Figures 7 and S15 are an indication of exactly this observation?! The fact that these only occur in early winter makes a lot of sense as well.

Lines 505-507: "... snow is a highly ... weeks." –> I am wondering whether it would

make sense to to have a thought experiment to check whether the magnitudes of the the changes involved fit this hypothesis. While Snow_overbar can be easily computed based on the snow properties and is independent of the sea-ice thickness and its radar freeboard, RF_overbar is not. It is a function of both, ice growth and snow load. An experiment one could think of is, e.g. an initially 80 cm thick ice floe with i) 5 cm snow depth and ii) 20 cm snow depth (and similar snow densities) grows at -30 degC over a month. What is the RF for both cases at the beginning and what is the change in RF over the month? Without further snow accumulation Snow_overbar remains constant. How would the change in RF be modified if one would add another 5 cm of snow in the middle of the month and again at the end of the month? My hypothesis is that RF_overbar and Ssnow_overbar are correlated well in case a thin to moderately deep, slowly increasing snow cover allows adequate ice thickening so that Snow_overbar increases over time but RF increases as well over time. Further, my hypothesis is that RF_overbar and Snow_overbar are not well correlated in case an already thick snow cover hampers adequate ice thickening while it further deepens over time. In that case Snow_overbar increase over time like in the above-mentioned example, but the RF increase by increasing SIT and hence sea-ice freeboard is counterbalanced by an increasing radar range such that the increase is considerably slower than in the first example or even zero.

Lines 513-517: "Freeze-up ... further study" –> I doubt that for the months you consider this is an issue. Melt ponds in the Central Arctic begin to freeze over in mid August, latest in September; hence any snow falling in October falls on solid ice. I suggest to delete this part.

Typos / Editoral remarks

Lines 5/6 " ... with the conventional method ... " –> I suggest to tie this better the usage of a snow climatology.

Line 91: I suggest to use a different letter for the dimensionless factor than rho

to avoid confusion with a density. Also "sigma_SIT_overbar" should possibly read "sigma_Snow_overbar"

Line 110: "radar speed" –> "radar wave speed" or "speed of the radar waves"

Line 118: If I am not mistaken, then Giles et al. (2008b) is purely dealing with the Antarctic, hence application of the W99 climatology appears to be unlikely.

Line 119: I agree that Eq. (2) contains the snow contribution to the sea-ice thickness; however, the term Snow_overbar is used in Eq. (4).

Line 134: "average half of the value" –> In order to allow a more fluent reading I suggest to write "about 50%" instead of "average half of the value" in Line 134. That way it will be easier to connect this finding with the SnowModel findings.

Line 139 and 140: "consistent" –> What is a consistent ice type in this context? Could it be that you wanted say "constant" or "unchanged"?

Line 187: "2005" –> "March 2005"

Lines 205: "assimilates reanalysis weather data" –> "is capable to assimilate meteorological data from different atmospheric reanalyses (see below)"

Lines 214: "results of reanalysis" –> "representation of the actual distribution of relevant meteorological parameters by atmospheric reanalyses"

Line 228: "Eq. (2)" –> Same issue as for Line 119 (see above)

Line 238: "The Central Arctic region exists above the latitudinal limit of the Envisat orbit" –> "The Central Arctic region is not sufficiently well observed by the Envisat radar altimeter (see Fig. 4)"

Line 255: "SnowModel's" –> I suggest to always keep the full name.

Line 257: "Having calculated ..." –> I suggest to refer to Figure 5 and the standard deviation shown therein. How about showing the detrended time series of RF_overbar?

Line 260: "between" –> "between detrended"

Line 278: "in for each" –> "for each"

Lines 286/287: "grouping by comparison to" –> "grouping in comparison to"

Lines 287-289: I suggest to add "radar" to all mentioning of "freeboard"

Figure 7 caption: For better readability of the figure and the text refering to it I suggest to somewhere re-introduce IAV here as the inter-annual variability. The last time IAV was used was in Figure 2. I cannot (a) and (b) in the Figure. I suggest to write "panel" instead of "figure in Line 293.

Line 294: "continuous" –> "contiguous"

Line 296: "was the" –> "was in the"

Line 306 "2002 - 2018": I suggest to add something like: "except for the Central Arctic: 2010-2018" in this caption as well as in all other figure captions where the Central Arctic region is shown along with results of the other regions.

Line 315: "Fig. 9" –> I suggest to refer to the panels highlighted in green.

Line 320: "declining" –> "negative"

Line 324: "mW99" –> Suggest to refer to Fig. 9 and the red panels

Line 325: "declining months to four" –> "months with a decline in SIT to four"

Line 326: "after 2006" –> I suggest to write "after 2003, except 2006"

Lines 333/334: "are only ones" –> "are the only ones"

Line 337: "Laptev Sea." –> I suggest to add "when using SnowModel-LG instead of mW99."

Line 340: "statistically significant months" –> When SIT is computed using SnowModel-LG?

Line 344: "snow" –> "show"

Figure 9, caption: I suggest to add the notion that y-axes of Central Arctic and East Siberian Sea differs from all other regions. I suggest to make "Where trends are ... superimposed" the second sentence of the caption. I suggest to add a note that unlike all other regions Central Arctic data are based solely on Cryosat-2 data.

Line 356: "Siberian there" –> "Siberian Sea there"

Line 404: "Chukchi" –> "Chukchi seas"

Line 405: "one month" –> "only one month"

Line 427: "raise" –> "rise"

Line 442: "diminishing show cover" –> I suggest to add: ", i.e. actual values of snow depth and SWE that are smaller than the climatological values"

Line 457: "EM" –> Has EM been explained already ...?

Line 489: "... truncated" –> Please re-phrase. It is not the radar altimetry time series that is truncated. There is a region where simply no measurements could be taken.

Line 511: "has longer to" –> "has longer time to"

Comments for the supplementary material:

Line 19: Please refer to Figure S1 here. Otherwise it is completely unclear how you ended up with the expression in Eq. (S8). You might want to replace the "=" by an "is approximated by ... as illustrated in Figure S1."

Line 23: "reformulated as" –> Which value is used for rho_water?

Figure S6, caption, line 3: "five" –> "four"

Figure S10, caption "... SnowModel-LG data." –> I suggest to add something like ... "expressed as total sea ice area of all grid cells falling into a specific SIT bin." In

addition: What is the bin size? What the bin borders? Since it includes the region "Central Arctic" this figure is based on years 2010-2018 only, correct?

Figure S11: There are two identical panels denoted (b). Please make a note in the caption about the differences in the y-axis range. Please make a note about the error bars. i.e. what these represent.

Figure S11, caption: "The SnowModel-LG contribution ..." –> While this statement is undoubtly correct also the aggregated marginal seas start low and end high. So why not also commenting on those regions? Again, I note that you need to provide the information whether all regions but the Central Arctic are based on the full period 2002-2018 or whether indeed all regions only used data from 2010-2018.

Figure S13: Again the notion about the smaller time period covered for region Central Arctic is missing.

---

## Referee Comment (RC2) · Anonymous Referee #2 · 13 Dec 2020

The manuscript argues that the snow climatology normally used when retrieving sea ice thickness from altimeter is missing trends and interannual variability This results in a statistical significant faster decline of Arctic sea ice in the Arctic marginal seas.

The link between the snow cover and the retrieved ice thickness is not new but the quantification is interesting and any progress towards understanding the snow cover is of importance. The use of Warren and modified Warren climatologies has been a issue for a while but there has not been any obvious alternative. I find this paper of interest to the community.

General notes Please be consistent and call sea ice the same. First example is on line 19 where it is mentioned as both sea ice and ice. I would prefer the first.

[Figure]

I would reconsider whether it is necessary to plot all panels for all areas and month in different figures. These become very small. Maybe it is better to show a few representative panels and put the rest in the supplementary material.

Minor comments

Line 10 and 11. It is true that knowledge about the polar climate is important for the polar climate, but I think that for this abstract it is a bit out of context to include stakeholders from Arctic shipping in the description. I would stop the sentence with mentioning the polar climate system (line 10).

Line 23 Agreed that thick ice has some of the properties mentioned, however thick ice do not make it easier to predict the ice cover. Assimilation of a correct ice thickness as oppose to a correct ice concentration has more memory and therefore the predictions are improved. Please rephrase

Line 34 ERS is mentioned here but its full name is mentioned on line 118. Please state full name here including the abbrivation and use the abbriviation in the rest of the text

Line 107 There is an issue with the reference. Henceforth W99?

Line 136: More a comment. It would be surprising if the variability of snow only depended on where the first year ice and the multi year ice was located.

Line 154 I would not start by describing why W99 is not mentioned.

Line 155: We instead compare. . . should include a reference to figure 3.

Line 168 remove one of "of". Typo.

Line 184: I would replace shaded with color coded.

Line 249 – 254 I think that the readability of this section can be improved if the flow of this section is improved.

Line 272. Is the Central Arctic for all ice types already mentioned in line 263. If this is

not the same point then please clarify.

Line 278 "in for each region in for each month." Should it be "one for each region and for each month. Please modify

Line 497. The las sentence should be reformulated. I think that a word is missing after positive in line 499.

Figure 1 W99 IAV values should be mentioned in figure text. If they are not used then remove the green bars

Figure 2 "Variability is displayed in a band where ice types typically fluctuates". Should this be High variability . . ..

Figure 5: I would reduce the number of panels and only show the ones that are commented on. The rest can go to the supplementary material. Details are very hard to see in these small panels.

Figure 6: The axis labels says meters but there are no ticks on the axis. I think that it should be added

Figure 7 a and b labels should be added. I can guess which are a and b but it should not be left to the reader to guess. In addition I would like to move the colorbar outside of the figures and enlarge it a bit, It should be commented why the fraction of total variance can extend beyond 0 and 100. For instance November central Arctic figure b (I suppose) exend from -18 to 118 (or something like that.

Figure 8 Should It be modified Warren? Labels say w99

---

## Author Comment (AC1) · 5 Feb 2021

We thank the anonymous referee for their useful comments and believe we have been able to address each of them.

Below we have copied their comments in blue and responded to each in red.

We would first like to first bring to the reviewer's attention a mistake made in the original manuscript. Due to a programming error we inadvertantly used radar freeboard data from a different product (that of Landy et al., 2020) in the winter of 2017/18. Because this product generally exhibits higher radar freeboard values than those used in the rest of the study due to a different retracking algorithm, we misidentified this winter at one point as 'a trend bucking year' for radar freeboards. We have now fixed this error and updated our statistics. This has had the following results:

- Regional declines in radar freeboard and resulting sea ice thickness are generally smoother.
- Negative trends in several regions are slightly increased.
- Negative trends are therefore more frequently statistically significant at the 5% level.
- Trends when calculated with SnowModel-LG in the 2002-2018 period are now in better agreement with those calculated from NESOSIM in the 2002-2015 period.

Despite these changes, the central thesis of our paper remains unchanged: the use of a snow product with regional variability and trends propagates into varaiblity and trends in regional sea ice thickness.

The manuscript argues that the snow climatology normally used when retrieving sea ice thickness from altimeter is missing trends and interannual variability This results in a statistical significant faster decline of Arctic sea ice in the Arctic marginal seas.

The link between the snow cover and the retrieved ice thickness is not new but the quantification is interesting and any progress towards understanding the snow cover is of importance. The use of Warren and modified Warren climatologies has been a issue for a while but there has not been any obvious alternative. I find this paper of interest to the community.

General notes

Please be consistent and call sea ice the same. First example is on line 19 where it is mentioned as both sea ice and ice. I would prefer the first.

We searched through the document and changed every mention of "ice" to "sea ice" where relevant. The only places where we have not done this is where the phrases "first year ice" and "multi-year ice" have been used, as we view these as standardised expressions.

I would reconsider whether it is necessary to plot all panels for all areas and month in different figures. These become very small. Maybe it is better to show a few representative panels and put the rest in the supplementary material.

We have removed three rows from Figure 6 in response to the specific comment concerning this issue. We have put a version of the figure with all regions in the supplement as suggested (and referenced it from the main text figure).

Minor Comments

Line 10 and 11. It is true that knowledge about the polar climate is important for the polar climate, but I think that for this abstract it is a bit out of context to include stakeholders from Arctic shipping in the description. I would stop the sentence with mentioning the polar climate system (line 10).

We have removed the words:

"as well as for stakeholders involved in Arctic shipping and natural resource extraction"

Line 23 Agreed that thick ice has some of the properties mentioned, however thick ice do not make it easier to predict the ice cover. Assimilation of a correct ice thickness as oppose to a correct ice concentration has more memory and therefore the predictions are improved. Please rephrase.

We have rephrased this section as follows:

"thick sea ice is far more likely to survive the melt season, increasing the average age of Arctic sea ice. Correct assimilation of ice thickness into models therefore offers opportunities for prediction of the sea ice state on seasonal timescales"

Line 34 ERS is mentioned here but its full name is mentioned on line 118. Please state full name here including the abbrivation and use the abbriviation in the rest of the text

We have now defined the acronym in the first instance and used just the acronym subsequently.

Line 107 There is an issue with the reference. Henceforth W99?

We have removed the abbreviation from this line and opted to define it when W99 is formally introduced in the Data Description section. The relevant part now reads:

All four groups utilize modified forms of the snow climatology assembled by Warren et al. (1999) from the observations of Soviet drifting stations between 1954 and 1991 (henceforth referred to as W99).

Line 136: More a comment. It would be surprising if the variability of snow only depended on where the first year ice and the multi year ice was located.

Yes, we agree! But the sea ice type distribution is the dominant determinant of snow variability in mW99.

Line 154 I would not start by describing why W99 is not mentioned.

We have since rearranged this section (on the advice of the other reviewer) and believe this to no-longer be an issue.

Line 155: We instead compare. . . should include a reference to figure 3.

We have added this reference.

Line 168 remove one of "of". Typo.

We have removed this typo

Line 184: I would replace shaded with color coded.

We have made this change.

Line 249 – 254 I think that the readability of this section can be improved if the flow of this section is improved.

We have comprehensively reworded this and added clarifying details concerning our approach. We note that on the advice of the other reviewer we have also repositioned this section.

Line 272. Is the Central Arctic for all ice types already mentioned in line 263. If this is not the same point then please clarify.

We were referring to the MYI fraction of the Central Arctic, and have added this detail to the sentence.

Line 278 "in for each region in for each month." Should it be "one for each region and for each month. Please modify

Yes this was a typo and we have changed it. The sentence now reads:

"the three components of $\sigma^2_{\overline{SIT}}$ for each region in each winter month"

Line 497. The last sentence should be reformulated. I think that a word is missing after positive in line 499.

We have reformulated it to read:

"The enhancement of declining trends where they exist is perhaps *of benefit* for these industries."

Figure 1 W99 IAV values should be mentioned in figure text. If they are not used then remove the green bars

We have reworded this part of section 4.1.1 to read:

" we find the snow variability introduced at a given point for mW99 (Fig. 3 blue bars) was on average about 50% of the value presented in W99 (Fig. 3, green bars)."

We note that this figure has been significantly repositioned on the advice of the other reviewer.

Figure 2 "Variability is displayed in a band where ice types typically fluctuates". Should this be High variability . . ..

Yes, we have made this change.

Figure 5: I would reduce the number of panels and only show the ones that are commented on. The rest can go to the supplementary material. Details are very hard to see in these small panels.

We have reduced the number of rows of this figure from eight to five by removing the rows corresponding to the Barents, Laptev and Kara Seas. The original figure has been moved to the supplement.

Figure 6: The axis labels says meters but there are no ticks on the axis. I think that it should be added

Because correlation statistics are not sensitive to the choice of axes, units or linear scalings of the values, we decided to not display axes ticks or labels and scale the axes to fit the rectangular panels of the figure. This decision was related to the issue highlighted in a previous comment about our figures being crowded. However we clearly should have stated this in our submission and we now have added the following text:

> "We note here that the correlation between the timeseries is dependent on their relative position to a linear regression. These correlation statistics are thus independent of the absolute magnitude of the values and any linear scaling of the axes. We therefore choose to present the correlations in Fig (7) without axes and scaled to the rectangular panels, so as to best show the relative positions of the points without extraneous numerical information."

The reviewer is correct also to point out that it is jarring to specify "(m)" as units without any axis ticks or tick-labels – we have therefore removed this from the axis labels (for reasons of unit-independence discussed above).

Figure 7 a and b labels should be added. I can guess which are a and b but it should not be left to the reader to guess.

We have now added these annotations

In addition I would like to move the colorbar outside of the figures and enlarge it a bit.

We have enlarged it and moved it up and outside the panels

It should be commented why the fraction of total variance can extend beyond 0 and 100. For instance November central Arctic figure b (I suppose) exend from -18 to 118 (or something like that.

We have now added some clarifying text:

[revised manuscript text omitted]

**Figure S4.** Basinwide trends in first year ice extent as a fraction of total extent from 2003-2018. Statistically significant trends exist in October (declining) and January (increasing). When trends of any significance are considered, all months show positive slopes barring October, which shows distinct decline. The October trend is due to later freeze-ups, the other positive trends fit in with established trends of increasing FYI dominance. Shaded regions represent the 95% confidence level for the linear regression.

[Figure]

**Figure S5.** Basinwide trends in mW99 SWE fields from 2003-2018. A statistically significant trend only exists in October, where SWE is increasing due to the increasing dominance of MYI in the month due to later freeze-ups. Shaded regions represent the 95% confidence level for the linear regression.

[Figure]

**Figure S6.** Detrended timeseries of spatially averaged snow contributions to sea ice thickness (Snow) by region from W99 (blue) and SnowModel-LG (red). Standard deviation values are displayed for SnowModel-LG (lower left, red), and mW99 (lower right, blue)

[Figure]

**Figure S7.** Detrended timeseries of spatially averaged snow contribution to sea ice thickness ($\overline{Snow}$) from W99 (blue) and SnowModel-LG (red) **over first year ice**. SnowModel-LG is significantly more variable from year to year than W99, which only varies due to shifting dominance of ice types. This increased variability propagates through to sea ice thickness, but is moderated by its covariance with radar freeboard variability. The standard deviations of the two timeseries are displayed in the lower corners of each panel.

[Figure]

**Figure S8.** Detrended timeseries of spatially averaged snow contribution to sea ice thickness ($\overline{Snow}$) from W99 (blue) and SnowModel-LG (red) **over multiyear ice** (MYI). SnowModel-LG is significantly more variable from year to year than W99, which only varies due to shifting dominance of ice types. This increased variability propagates through to sea ice thickness, but is moderated by its covariance with radar freeboard variability. A substantial number of data points are missing from some panels - these absences reflect months where no MYI is present in the relevant region. The standard deviations of the two timeseries are displayed in the lower corners of each panel.

[Figure]

**Figure S9.** FYI correlations between radar freeboard and snow contributions to sea ice thickness, where the snow contribution is calculated using SnowModel-LG. All statistically significant correlations are positive (i.e. years with more snow exhibit higher radar freeboards). A persistent, positive correlation exists in the Central Arctic and the East Siberian Sea in the last five months of winter. The Barents and Kara Seas both exhibit significant correlations in the last two months of winter. The Beaufort sea exhibits no months of statistically significant correlation between radar freeboard and snow contributions.

[Figure]

**Figure S10.** MYI correlations between radar freeboard and snow contributions to sea ice thickness, where the snow contribution is calculated using SnowModel-LG. Fewer correlations exist for MYI than for FYI. The Central Arctic and Chukchi Sea exhibit no correlations between snow and radar freeboard contributions.

[Figure]

**Figure S11.** Regional IAV displayed by ice type. MYI represented by orange points, FYI represented by purple. When averaging over the growth season in a given region, MYI is more variable in all the marginal seas.

[Figure]

**Figure S12.** Detrended timeseries of spatially averaged sea ice thickness ($\overline{SIT}$) by region from W99 (blue) and SnowModel-LG (red) for **all ice types**. Standard deviation values are displayed for SnowModel-LG (lower left, red), and mW99 (lower right, blue).

[Figure]

**Figure S13.** 2010-2018 basin-wide sea ice thickness distribution calculated using both mW99 and SnowModel-LG data expressed as total sea ice area of all grid cells falling into a specific SIT bin. Bin size is 5 cm. Shaded areas represent the area constituted by the Central Arctic.

[Figure]

**Figure S14.** Seasonal evolution of (a) snow thickness and (b) sea ice thickness by region. All regions calculated over 2002-2018 with the exception of the Central Arctic, which is 2010-2018. Note different y-axis scales for Central Arctic panels. 'Error bars' represent the one standard-deviation range either side of the mean value for the timeseries. The SnowModel-LG contribution starts lower but ends higher in the Central Arctic, the region that dominates Pan-Arctic statistics. This is also true for the Marginal Seas grouping, but not necessarily true for the individual constituent regions. This corresponds to faster thickness increase than would be calculated with W99.

[Figure]

**Figure S15.** Interannual variability of SnowModel-LG contribution to $\sigma^2_{\overline{SIT}}$ ($\sigma^2_{\overline{Snow}}$) when forced by two different reanalysis data sets. MERRA2 (orange) and ERA5 (green) produce very similar variability.

[Figure]

**Figure S16.** Trends in sea ice thickness (2002-2018) by region, when calculated using SnowModel-LG runs using two different sources of reanalysis (ERA5, Purple; MERRA2, Orange). Panels are framed with green where statistically significant trends exist independent of reanalysis choice. Purple (orange) frames represent month/region pairs where statistically significant trends are only present with ERA5 (MERRA2). Slope values are given where significant in the lower corners. All significant trends in the marginal seas are negative, all significant trends in the Central Arctic are positive. In the Central Arctic, two of the four statistically significant increasing trends are only evident with ERA5 reanalysis. In the Marginal Seas, the decline in some months is only statistically significant with MERRA2.

[Figure]

**Figure S17.** Trends in snow contribution to sea ice thickness ($\overline{Snow}$; 2002-2018) by region, when calculated using SnowModel-LG runs using two different sources of reanalysis (ERA5, Purple; MERRA2, Orange). Panels are framed with green where statistically significant trends exist independent of reanalysis choice. Purple (orange) frames represent month/region pairs where statistically significant trends are only present with ERA5 (MERRA2). Slope values are given where significant in the lower corners.

[Figure]

**Figure S18.** Interannual variability of NESOSIM data's contribution to SIT, shown as (a) absolute contribution to SIT variability, and (b) relative contribution. Variability from snow is of a similar magnitude to that of SnowModel-LG, although regional differences exist between the corresponding plots, particularly in the Barents Sea. As well as differences in the snow accumulation scheme, the two data sets differ in spatial resolution and the timespan over which they are analysed.

[Figure]

**Figure S19.** Timeseries of the thickness contributions of radar freeboards ($\overline{RF}$) and snow ($\overline{Snow}$) over all ice types. Orange framed boxes indicate statistically significant decline in both $\overline{RF}$ and $\overline{Snow}$. The red framed box indicates statistically significant decline in $\overline{Snow}$ only. No boxes feature a statistically significant decline in $\overline{RF}$ without a concomitant decline in $\overline{Snow}$. All statistically significant trends in both $\overline{Snow}$ and $\overline{RF}$ are negative.

---

## Author Comment (AC2) · 5 Feb 2021

We would like to thank Dr. Kern for his thoughtful and constructive review; we believe that his feedback will undoubtedly improve the quality of our manuscript.

We have appended an ammended manuscript to this document which illustrates the changes we have made to our submission in response to the feedback. Below we quote Dr. Kern's comments in blue, and our responses follow in red.

We would first like to first bring to the reviewer's attention a mistake made in the original manuscript. Due to a programming error we inadvertantly used radar freeboard data from a different product (that of Landy et al., 2020) in the winter of 2017/18. Because this product generally exhibits higher radar freeboard values than those used in the rest of the study due to a different retracking algorithm, we misidentified this winter at one point as 'a trend bucking year' for radar freeboards. We have now fixed this error and updated our statistics. This has had the following results:

- Regional declines in radar freeboard and resulting sea ice thickness are generally smoother.
- Negative trends in several regions are slightly increased.
- Negative trends are therefore more frequently statistically significant at the 5% level.
- Trends when calculated with SnowModel-LG in the 2002-2018 period are now in better agreement with those calculated from NESOSIM in the 2002-2015 period.

Despite these changes, the central thesis of our paper remains unchanged: the use of a snow product with regional variability and trends propagates into variability and trends in regional sea ice thickness.

Summary: This very interesting paper illustrates the potential improvement in the credibility of trends in and inter-annual / intra-basin variations of Arctic sea-ice thickness estimates from satellite radar altimetry. This is achieved by a comprehensive intercomparison of the contribution of snow on sea ice on the retrieval of sea-ice thickness from radar freeboard when using the Warren et al. (1999) snow climatology on the one hand and a physical model for snow properties driven by atmospheric reanalyses' precipitation and other relevant meteorological parameters on the other hand. As expected, the inter-annual variability of the snow contributions based on the model data is considerably larger than the one based on the Warren et al. (1999) data. The paper further convincingly demonstrates that the more realistic inter-annual variability and spatio-temporal development of the Arctic Ocean sea-ice thickness.

The paper is generally well written and will have considerable impact on the scientific community. It would benefit from some re-organization (see GC1). It is furthermore quite light when it comes to descriptions of data and methodologies used (GC2). Currently, one would not be able to re-produce the work done. The inclusion of Kara and Barents Sea I find quite a hypothetical move based on the data availability and suggest to consider removing those from the analysis (GC3). Finally, there is a number of open points to discuss when it comes to the illustration and interpretation of the results presented. In the following you will find my list of general comments (GC), specific comments and some suggestions to mitigate typos and editoral issues - all for the main manuscript - followed by a short list of things I found worth to consider in the supplementary material.

Title: While your main conclusion supports the title in general, it is in some way misleading. The main focus of the paper is on the illustration that a snow depth climatology is not well suited to compute credible trends in sea-ice thickness estimates derived from

satellite altimetry with such a snow depth as input for the freeboard to thickness conversion. In your paper, this is illustrated by usage of data from a numerical model which has experienced limited validation. Hence, albeit the improvement using these model data is obvious it is not necessarily the truth either. Hence, instead of formulating the title as a fact I suggest to include points of the above-stated.

In response to this feedback we would append a clarifying clause, so that the title reads:

'Faster decline and higher variability in the sea ice thickness of the marginal Arctic seas *when accounting for dynamic snow cover*'

GC1: I strongly recommend to re-organize the paper. Most of the explanations / motivations given in the subsections 1.1 and 1.2 are tied relatively close to Section 3 and should be combined with that section. In addition, subsections 1.1 and 1.2 refer to data and regions denoted in Section 2. Hence: Remove 1.1 and 1.2 and put it into Section 3. Let Section 2 start right behind the "true" introduction. That way the data sets used in 1.1 and 1.2 would be introduced adequately beforehand which eases reading and which reduces the number of open questions.

We have rearranged these sections accordingly. Section 1.1 (on our method of separating the impacts of snow and radar freeboard data on thickness determination) has been moved to the methods section (Section 3). We agree with the reviewer that our illustration of the limitations of W99 in Sects 1.2.1 & 1.2.2 would have been better placed after the data description (Sect 2). Rather than put this in the Methods section (Sect 3), we have moved it to the beginning of our Results section (Sect 4). We hope this is satisfactory to the reviewer, but if not we will of course reconsider his original suggestion of Sect 3.

GC2: Both, the description of the data used as well as of the methodologies used lack some clarity and/or do not contain all information required. One good example: The ESA-CCI radar freeboard data set used comprises data of two different satellites with some overlap. It is not clear from the description in the data how long the Envisat and how long the Cryosat-2 part of the data used is - plus a motivation of the choice made - plus a discussion about the biases between the radar freeboards of these two satellites, which have a different sign based upon the region. Some of the descriptions also appear to contain errors which ask for re-phrasing.

We agree that not specifying the transition point from Envisat to CryoSat-2 radar freeboard data was an oversight. We would add the following clarifying information into the "Radar Freeboard Data" subsection:

CS2 carries a delay-Doppler altimeter that significantly enhances along-track resolution by creating a synthetic aperture. For this reason as well as its higher latitudinal limit, we used CS2 radar freeboard measurements over Envisat's during the period when the missions overlapped (November 2010 - March 2012).

We have also made changes to our data description and discussion sections with regard to the potential effect of biases between the radar freeboard measurements of the two satellites. To "Data Description" (Sect 2.2) we have added:

To create a radar freeboard product that is consistent between the Envisat and CS2 missions, Envisat returns are retracked using a variable threshold retracking algorithm. This variable threshold is calculated from the strength of the surface backscatter and the width of the leading edge of the return waveform such that the inter-mission bias is minimised (Paul et al., 2018). The results are comprehensively analysed in the Product

Validation & Intercomparison Report (ESA, 2018). One key finding of this report is that while Envisat radar freeboards are calculated so as to match CS2 freeboards during the period of overlap over the whole Arctic basin, there are biases over ice types. In particular, Envisat ice freeboards (not radar freeboards) are biased 2-3 cm low (relative to CS2) in areas dominated by MYI, and 2-3 cm high in areas dominated by FYI. We discuss the implications of these biases in Sect. (5.3).

To our Discussion section, we have added a subsection "Inter-Mission Bias between Envisat and CryoSat-2" (5.3). This reads:

An extensive validation exercise for the merged products indicated that although Envisat radar freeboards match well with CS2 freeboards in the Arctic overall, some biases do exist over specific ice types (ESA, 2018). In particular, analysis of the inter-mission overlap period indicates that Envisat freeboards were biased low (relative to CS2) in areas dominated by MYI, and high in areas dominated by FYI.

We first make the point that this will have a relatively minimal effect on our findings regarding interannual variability, as  $\overline{\text{Snow}}$  is unaffected by this and  $\sigma^2_{RF}$  is likely relatively independent of the absolute magnitude of  $\overline{\text{RF}}$ .

With regard to trends, if Envisat radar freeboards (and thus  $\overline{\text{RF}}$ ) are in fact biased high over FYI between 2002-2010 (relative to CS2), then the total trend in many regions dominated by FYI could potentially be smaller than calculated in this manuscript.

We do however add that our findings regarding the impact of declining Snow is unaffected by any inter-mission bias in RF. Because the trend in SIT is determined by both Snow & RF, the trend in SIT will always be more negative when calculated with downward trending data for Snow.

GC3: The overall credibility of the paper would benefit from a more critical consideration of the application area of the Warren et al. (1999) climatology. Sampling density, number of observations, as well as the distribution of the snow depth observations over time combined with the usage of a polynomial fit limits the usefulness of these observations in the regions Kara Sea and Barents Sea. One good solution would be to omit these regions.

Following the suggestion of the reviewer we analysed the original positional data (found within the meteorological observations) from drifting stations NP3 – NP31. This was all the data available to us, and was supplied previously in a personal communication by the NSIDC.

After plotting the tracks of these 27 drifting stations, we counted the number of stations that visited each region in each month:

We note in the above figure that repeat visits by stations on the same or consecutive years do not add to the tally. For example, NP22 lasted for four years and visited the same regions in the same months on consecutive years. However it then became apparent that some stations were not making snow measurements during some regional 'visits', and this should be included in the consideration of the sampling density as suggested by the reviewer. Instead we identified all distinct dates on which snow stake data was gathered by each NP station. We then cross-referenced this with the positional data found within the met data, to break down the number of distinct stake-measurement-days in each region by month. We believe this is a suitable metric for the spatial sampling of the drifting stations.